# IgStrand: A universal residue numbering scheme for the immunoglobulin-fold (Ig-fold) to study Ig-proteomes and Ig-interactomes

Caesar Tawfeeq[1], Jiyao Wang[2], Umesh Khaniya[3], Thomas Madej[2], James Song[2], Ravinder Abrol[1]*, Philippe Youkharibache[3]*

1 Department of Chemistry and Biochemistry, California State University Northridge, Northridge, California, United States of America, 2 National Center for Biotechnology Information, National Library of Medicine, National Institutes of Health, Bethesda, Maryland, United States of America, 3 Cancer Data Science Laboratory, Center for Cancer Research, National Cancer Institute, National Institutes of Health, Bethesda, Maryland, United States of America

* abrol@csun.edu (RA); philippe.youkharibache@nih.gov (PY)

## Abstract

The Immunoglobulin fold (Ig-fold) is found in proteins from all domains of life and represents the most populous fold in the human genome, with current estimates ranging from 2 to 3% of protein coding regions. That proportion is much higher in the surfaceome where Ig and Ig-like domains orchestrate cell-cell recognition, adhesion and signaling. The ability of Ig-domains to reliably fold and self-assemble through highly specific interfaces represents a remarkable property of these domains, making them key elements of molecular interaction systems: the immune system, the nervous system, the vascular system and the muscular system. We define a universal residue numbering scheme, common to all domains sharing the Ig-fold in order to study the wide spectrum of Ig-domain variants constituting the Ig-proteome and Ig-Ig interactomes at the heart of these *systems*. The "IgStrand numbering scheme" enables the identification of Ig structural proteomes and interactomes in and between any species, and comparative structural, functional, and evolutionary analyses. We review how Ig-domains are classified today as topological and structural variants and highlight the *"Ig-fold irreducible structural signature"* shared by all of them. The IgStrand numbering scheme lays the foundation for the systematic annotation of structural proteomes by detecting and accurately labeling Ig-, Ig-like and Ig-extended domains in proteins, which are poorly annotated in current databases and opens the door to accurate machine learning. Importantly, it sheds light on the robust *Ig protein folding algorithm* used by nature to form beta sandwich supersecondary structures. The numbering scheme powers an algorithm implemented in the interactive structural analysis software iCn3D to systematically recognize Ig-domains, annotate them and perform detailed analyses comparing any domain sharing the Ig-fold in sequence, topology and structure, regardless of their diverse topologies or origin. The scheme provides a robust fold detection and labeling mechanism that reveals unsuspected structural homologies among protein structures beyond currently identified Ig- and Ig-like domain variants. Indeed, multiple folds classified independently contain a common structural signature, in particular jelly-rolls. Examples of folds that harbor an "Ig-extended" architecture are given. Applications in protein engineering around the Ig-architecture are straightforward based on the universal numbering.

**Data availability statement:** All software code is available in github (https://github.com/NCBI/iCn3D), also as indicated with specific links in the text. All other relevant data are in the manuscript and its supporting information file.

**Funding:** This work was supported in parts by the Intramural Research Program of the National Institutes of Health (www.nih.gov) at the National Cancer Institute for PY and at the National Center for Biotechnology Information of the National Library of Medicine for JW, TM, JS; and by the National Science Foundation (grant 2346274) (www.nsf.gov) for RA. The funders had no role in study design, data collection and analysis, decision to publish, or preparation of the manuscript.

**Competing interests:** The authors have declared that no competing interests exist.

## Author summary

The Immunoglobulin fold (Ig-fold) is a highly conserved protein architecture that has diversified extensively throughout evolution to provide a significant number of Ig-domain types with variable topologies in all life forms. Primarily known for its role in the vertebrate immune system and in particular for its presence in the structure of antibodies, the Ig-domains are found in a myriad of proteins involved in multiple diverse biological/structural functions within the immune, nervous, vascular, and muscular systems, where they mediate neural development, cell-cell recognition, cell adhesion and signaling. Structurally, the Ig-fold consists of 70–110 amino acids that form two beta sheets stabilized by hydrogen bonds facing each other in a sandwich configuration. This architecture provides stability and tremendous adaptability, making the Ig-fold a fundamental building block in many protein families across species, from bacteria to vertebrates. The Ig-fold is most abundant in the human genome, accounting for at least 2–3% of protein-coding regions, which is an underestimate as no reliable method exists to date that can identify all Ig domain variants with precision. To better identify and to study Ig-domain variants and most importantly their specific interactions behind biological functions, the "IgStrand universal residue numbering scheme" was developed. It enables the annotation of the diverse Ig-domain topological variants down to the residue level and their interactions, allowing detailed comparative structural, functional, and evolutionary analyses. This deconstruction of all known Ig-domain topological variant tertiary and quaternary structures will assist in the rational design from scratch of Ig-based therapeutics from single-domain antibodies to cellular therapies against any antigen of interest.

## Introduction

### Proteins containing the Ig fold in the Human proteome

Protein domains sharing the Ig-fold are often referred to as belonging to the Ig Superfamily (IgSF). Ig-domains have been identified as **the most populous superfamily in the human genome, with an initial estimate of over 2% of coding genes** [1]. The Ig-fold is found in all domains of life, yet nowhere as pervasively as in vertebrates where it is essential in multiple intracellular, extracellular *and intercellular* communication functions. As such, it is a key element of the immune system, the nervous system, the vascular system and the muscular system. **Fig 1** shows a sample of protein families containing Ig domains, yet it would take a very large encyclopedia to cover the reach of Ig-fold in all life forms or even just in humans. To date, it has been found in numerous proteins in a myriad of topological variants, in all subcellular compartments from the nucleus to the cytoplasm to extracellular regions. It can be considered as **a wunderkind of evolution**. The question we are addressing is: **what makes it such a superfold?** How can we uncover the reasons for its success, powering so many diverse biological functions? How can we study its folding across so many variant forms, so many functions, in so many unicellular and multicellular species, in so many diverse tertiary chain arrangements or quaternary molecular complexes?

Ig-like folds have been found in viruses, bacteria, archaea and eukaryotes [2–5], and in that latter kingdom from unicellular to multicellular organisms [6]. They can be found in the nucleus or the nucleus surface (Lamin, POM210, P53, NFκB, PTEN), in the cytoplasm (β-Arrestins, PKC, PI3K, PLC, Synaptotagmin), and on the cell surface (CD4, CD8, CTLA-4, TCR, NCAM, ICAM), and even as secretable proteins such as antibodies and transthyretin. It

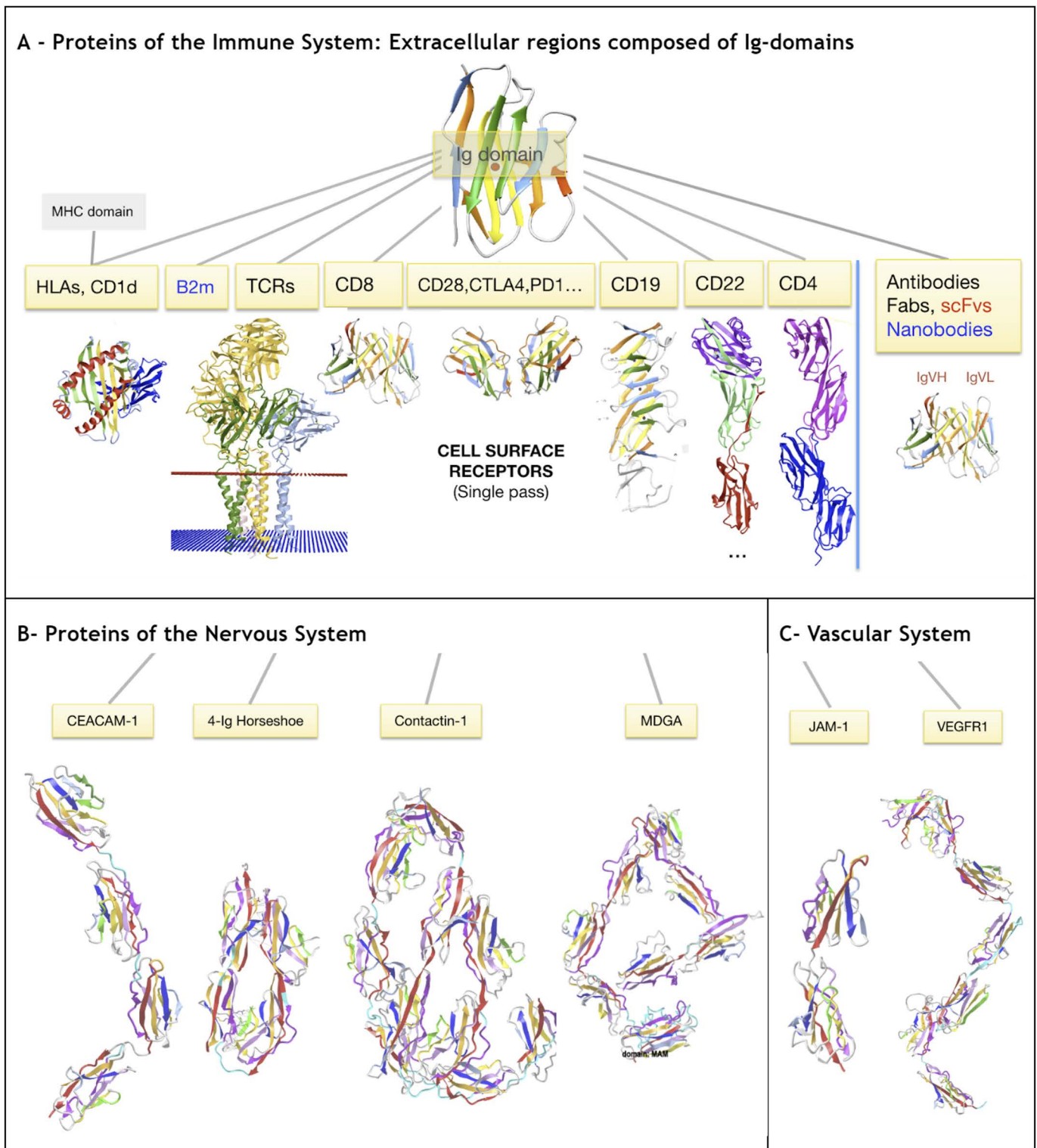

**Fig 1. Diverse proteins containing a variable number of Ig-like domains chained in tandem. A) immune system proteins:** TCRs, MHCs, antibodies and many other cell surface receptors extracellular regions contain Ig-domains (CD8, CD4, CD22, CD28, PD1, CTLA4 etc.). **B) Nervous system** proteins: CEA-CAMs, contactins, that like sidekick or DSCAM proteins contain a 4-Ig horseshoe "superdomain" at the N-terminus (see section on Ig-Ig interfaces). MDGA proteins (MAM domain-containing glycosylphosphatidylinositol anchor) contain 7 Ig-domains followed by a MAM domain that has a jelly roll fold with similarities with the Ig-fold. C) Vascular **system** proteins: JAM proteins contain 2 Ig-domains in tandem, and VEGFR act as receptors for VEGF (Vascular

endothelial growth factor) proteins. 3D models can be visualized in iCn3D links: Contactin1 https://structure.ncbi.nlm.nih.gov/icn3d/share.html?imRwYo-JKmCpCaXYm7&t=Q12860; CEACAM1 https://structure.ncbi.nlm.nih.gov/icn3d/share.html?g89JayoAbbjN7Yg18&t=P13688; VCAM1 https://structure.ncbi.nlm.nih.gov/icn3d/share.html?9edsvYBJ43ZyPv9j8&t=P19320; VEGFR1 https://structure.ncbi.nlm.nih.gov/icn3d/share.html?SbL83DUuwccu-JWtz8&t=Q8QHL3; JAM1 https://structure.ncbi.nlm.nih.gov/icn3d/share.html?6mGp5wdHxcxV4Cv98&t=1NBQ.

is fair to say that the Ig-fold is a wunderkind of structural evolution, irrespective of divergent or convergent evolutionary paths. To aim at understanding key elements governing the folding of the Ig-fold in its multiple domain variants, tertiary architectures of chains composed of Ig domains and their quaternary assemblies, **we seek a way to compare and analyze all protein domains that share the Ig-fold, their tertiary chains and quaternary assemblies.**

Immunoglobulin (Ig) or Ig-like domains expressed on the surface of cells are of particular interest, as they orchestrate cell-cell recognition and adhesion to trigger further cell signaling and gene expression through receptor/ligand interactions. A remarkable aspect of the Ig-based cell communication system is the symmetry between receptors and ligands, *both* made of Ig- or Ig-like domains, making the Ig-fold a unique and special element of the immune system from which it inherits its name [2,7–12]. Similar Ig-based communication is present in the nervous system [13–20], the vascular system [21–23], and the muscular systems [24–29].

This systemic omnipresence in complex organisms *underscores the importance of the Ig-fold in health and disease.* Indeed, a myriad of diseases are rooted in Ig-domain dysfunctions that range from cancer and autoimmune diseases to muscular diseases [30,31], vascular diseases [22], and amyloidosis [32,33]. As science and medicine start exploring the immune-muscle systems interface, the Ig-domains appear central, with skeletal muscles expressing immunomodulatory molecules such as ICAM-1, ICOSL, CD80 or PD-L1 [34,35]. It is also the case in the immune-nervous systems interface [36–42] and the neuroimmune cardiovascular interface where, for example, the parallel role of semaphorins in neurons and immune cells has become an intense field of research [43–45].

Many cell surface proteins of the nervous system are composed of Ig-domains chained in tandem in their extracellular region and are involved in cell adhesion (CAMs: NCAM, CEACAM, L1CAM, MDGA, OBCAM, SYnCAM, etc.). Some, such as contactins, Sidekick proteins, DSCAM and others, form a horseshoe "superdomain" with the four N-terminal Ig domains of the extracellular Ig-chain, and contain FN3 domains in the C-terminal membrane proximal region of the extracellular Ig-chain. Cell adhesion Molecules in the vascular system are also built out of Ig-domains (ICAM, VCAM, PECAM, JAM, etc.). For example, Vascular Endothelial Growth Factor Receptor (VEGFR) is a Receptor-type Tyrosine Kinase (RTK) consisting of a ligand-binding region with seven Ig-like domains as its extracellular region. In vertebrates it plays essential roles in the regulation of angiogenesis and lymphangiogenesis. In the muscular system Obscurin and Titin [27] are giant macromolecules composed for the latter of over 200 Ig-like domains.

*To enable the systematic and parallel study of **the Ig-folded elements** of the molecular and cellular systems they articulate, we need a means to bring them into a common frame of reference in sequence, topology and structure.*

## The Ig fold

**The classical Ig-domains: IgV, IgI, IgC1 and IgC2.** The Immunoglobulin fold is common to multiple domain variants in topology. It is usually described as a beta sheet sandwich barrel and can contain from 7 to 9 strands (A, B, C, [C', C,"] D, E, F, G) for the main variants. The Ig-fold can even have additional strands in cases we call Ig-extended. As we shall see, there are multiple variants in topology and structure, but four main variant types, named V-set (IgV),

C1-set (IgC1), C2-set (IgC2), and I-set (IgI) (see **Fig 2**) [46–48] have been studied extensively in the human immune system in particular. Among them, and since its original structural determination [49–54], the Immunoglobulin variable domain has been the most studied, as it is found at the N terminus of antibodies in their Heavy and Light chains (VH and VL

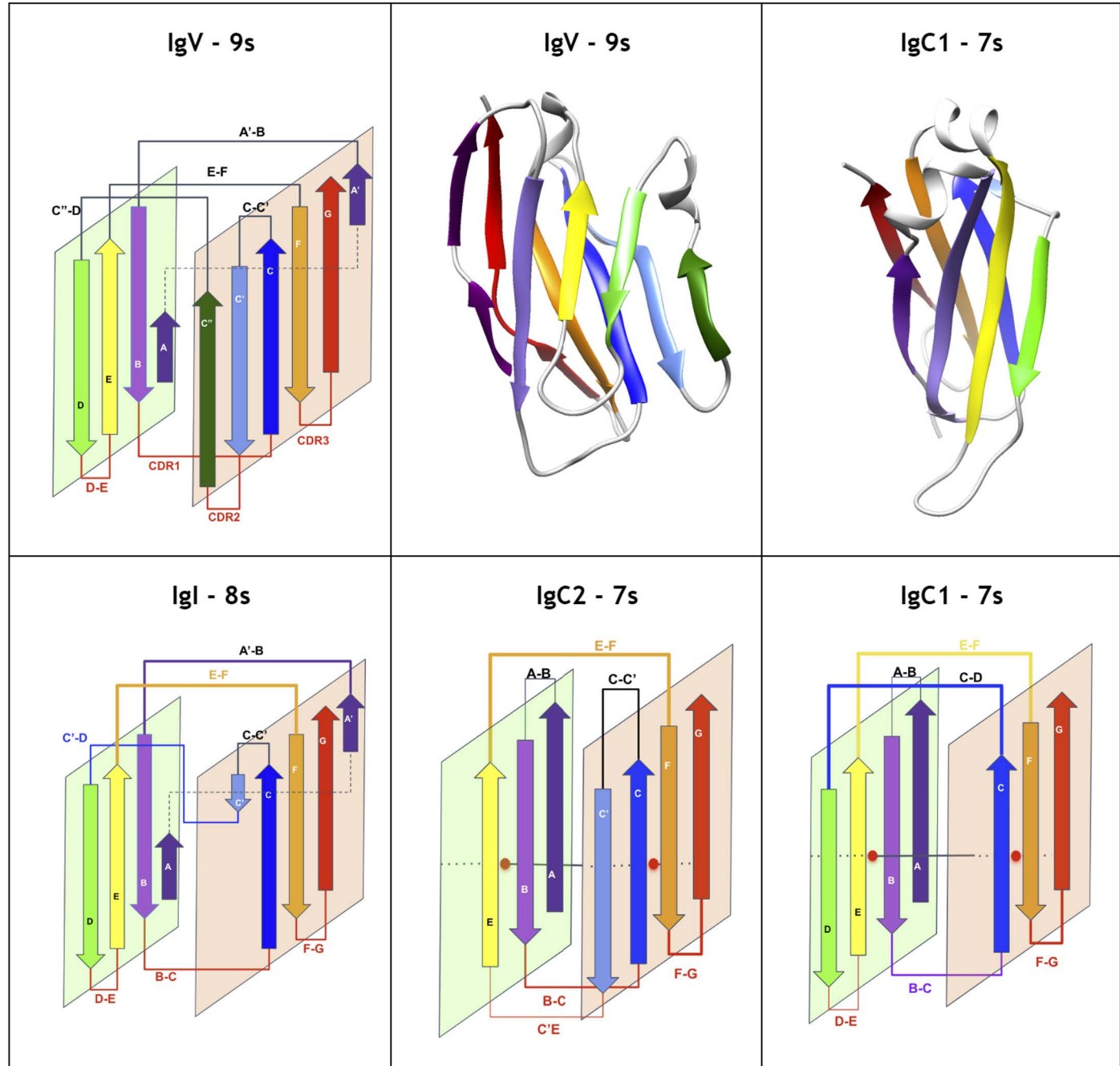

**Fig 2. Canonical Ig domain variants: Schematic and Ribbon representations. The IgV domain** contains 9strands (AA')BCC'C''DEFG (9s) according to the nomenclature as displayed, The IgI domain variant contains 8 strands (AA')BCC'DEFG, IgC1 and IgC2 contain both 7 strand, ABCDEFG and ABCC'EFG resp.**.** The A strand splits as A and A' between the two sheets of the beta sandwich in IgI and IgV domains, as displayed. Some IgV domains can exhibit A' strand only. The IgC2 can also present an A strand split. (see text for details).

domains) and on T-cell Receptors (TCRs), where it is responsible for binding antigens and neoantigens respectively, hence its immense importance in immunity. In addition, immune cells are harnessed with a multitude of cell surface receptors composed of Ig domains chained in tandem starting with IgV domains at their N termini followed by IgC2 or IgC1 domains in most cases [11,55–58]. Beyond the four classical variants, sub-variants have also been further classified (V1, V2, I1, I2, I3) to account for small structural variations. However, the Ig-fold can be seen in many more topological, structural and functional variants.

I-set domains are commonly found in various cell adhesion molecules, such as vascular cell adhesion molecule (VCAM), intercellular adhesion molecule (ICAM), neural cell adhesion molecule (NCAM), mucosal addressin cell adhesion molecule (MADCAM), and junction adhesion molecules (JAM) [47]. Additionally, the I-set domains are present in diverse protein families, including tyrosine kinase receptors, the hemolymph protein hemolin, muscle proteins like titin, telokin, and twitchin, the neuronal adhesion molecule axonin-1, the signaling molecule semaphorin 4D involved in axonal guidance, immune function, and angiogenesis. V-set and I-set Ig domains are particularly important in forming ectodomains of cell surface receptors, with secondary and tertiary structures that closely resemble each other, and phylogenetic analyses frequently group them together. The striking similarity between V-set and I-set Ig domains shows their shared evolutionary heritage and suggests conserved function in cell surface receptors.

**Sequence patterns of classical Ig-domains.** The four classical topological variants of the Ig-fold (IgV, IgC1, IgC2, and IgI) are highly conserved in sequence and structure. They have been extensively studied and classified and are the object of a curated database [59,60] (Fig A in S1 Text). **Fig 3** presents a graphical view of their strand-aligned sequence patterns, where one can appreciate the level of conservation of CCW(L) key residues in the core BC|EF strands [61,62] despite topological variations. Fig 3 captures the most commonly observed strands, however, it doesn't capture the diversity of many of the Ig domains. For example, IgV domains may present a split A/A' strand, or just an A' strand as in CD4 or only an A strand as in ICOS. Some IgV-domains may be missing the C" strand altogether as in PD-L2 or VNAR domains in sharks. **Fig 3** also presents FN3 and Cadherins sequence logos (aligned to the four classical Ig topologies) representing two major Ig-fold variants with different sequence patterns, except for a conserved Tyrosine in the F strand (called the Y corner, see later discussion of this figure) and other less conserved residues present across the BCEFG strands. Structural conservation despite a lack of sequence conservation is a hallmark of the Ig-fold variants in topology and structure, which can point to both divergent and convergent evolution [63]. Despite the strand topologies observed for the four canonical Ig-domains that have been used extensively in annotating Ig-domains [59,60] (see Fig A in S1 Text), some observed Ig-domains evade the classification using the V, I, C1, C2 types. Indeed, many more Ig-fold variants have been identified, such as the IgE set (early Ig domains) and domains that share the fold but are significantly diverse in sequence, topology and structure.

**Known topo-structural variants of the Ig-fold beyond the classical Ig-domains.** The first immunoglobulin variable domain homodimer X-ray structure, the Bence-Jones protein [64,65], was part of the very first set of three structures deposited to the PDB [PDBid 1REI]. To date, the protein data bank (PDB) provides 16,661 files containing Ig-like structures (as determined by IgStrand) representing 7.5% or the largest body of experimentally determined structures available on any domain (https://www.rcsb.org/) with over 25,000 Ig|Ig domain interfaces represented. The largest portion of experimental structures contain antibodies exhibiting the canonical VH:VL and CH1:CL interfaces. But, overall, this body of data covers multiple types of different Ig domain quaternary interfaces. Over the years, the growing body of experimental structures in the PDB has shown an ever-growing set of Ig-domain variants

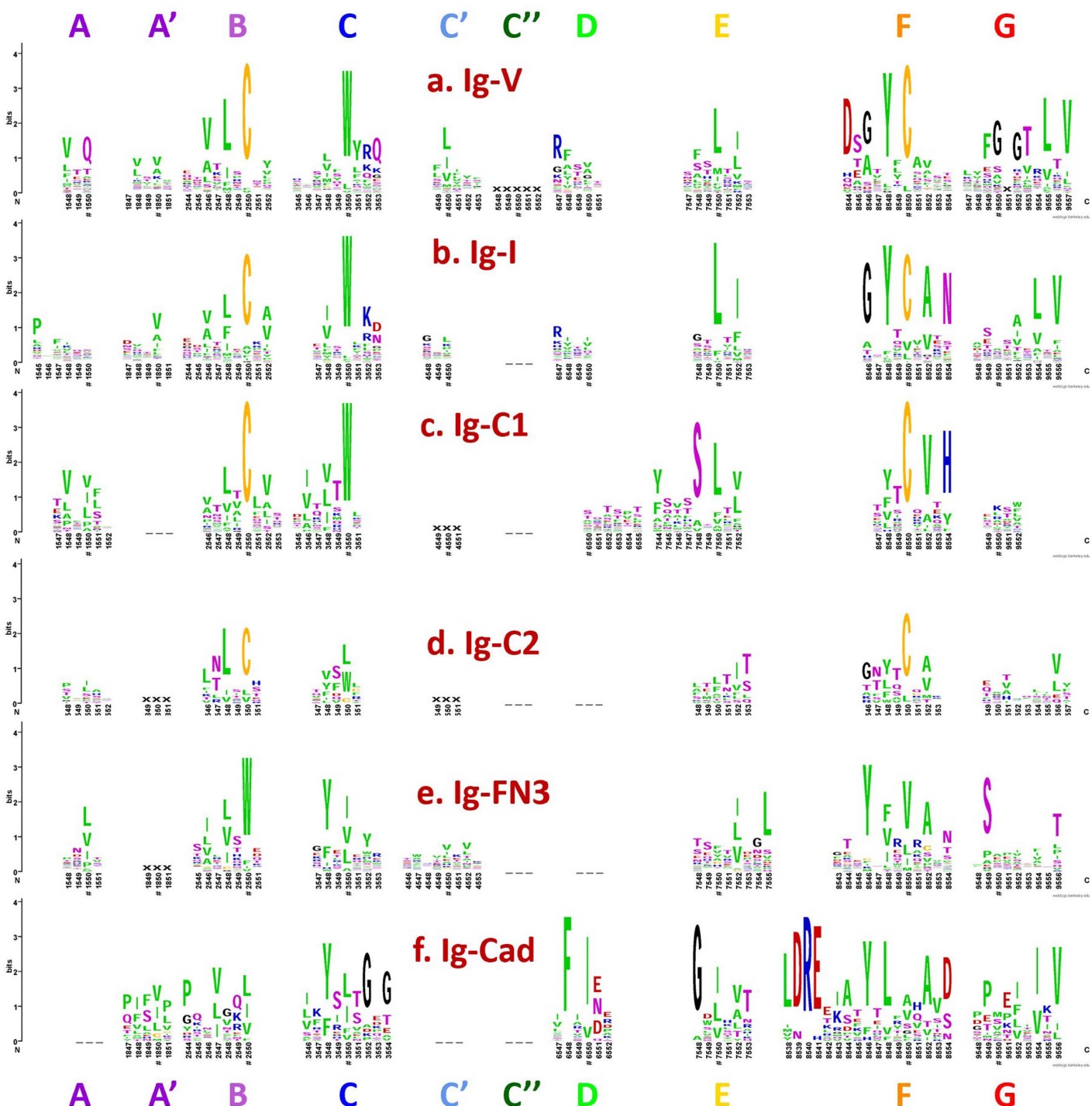

**Fig 3. Sequence patterns of Ig-domain variants in WebLogo format.** The anchor residues in each strand are marked by a "#", which are numbered xx50 as described in the Results and Discussion section. a) The IgV-set with most commonly an (AA')BCC'C"DEFG topology (9 strands). b) The IgI-set presents a (AA') BCC'DEFG topology (8 strands) highly similar to IgV domains. c) The IgC1-set exhibits an ABCDEFG topology (7 strands), with usually two hallmarks: a straight A strand, even if some strand breaks can be observed, and a D strand. Some IgC1 domains can also exhibit a small non-conserved C' strand (denoted by XXX) as in IgI or IgC2 domains. d) The IgC2-set exhibits an ABCC'EFG topology (7 strands). IgC2 domain can exhibit an A/A' strand split with a non-conserved A' strand (denoted by XXX) as in LILRs. It also has a C' strand with no sequence conservation (denoted by XXX). IgC2 differs from IgC1 domains in not presenting a D strand. The first four canonical Ig-domains IgV, IgI, IgC1 and IgC2 present high similarity in sequence, especially the Cysteines in strands B and F forming a Cys-Cys bridge and a Tryptophan in strand C flanking the Cys-bridge. Each possess shared specific sequence patterns, for example the Tyrosine corner in strand **F** (see later in text) or some specific to each type such as R and N residues in IgI strand D and F respectively, residues Q, RF and D in IgV's strand A, D and F respectively,

etc. One should note that the G-strand sequence pattern in the WebLogo results from an over representation of antibody domains in the dataset. e) The **Ig-FN3 set** presents an ABCC'EFG topology (7 strands). Like any other Ig-domain variant some FN3 domains can exhibit an A/A' strand split. f) The **Ig-Cadherin set** exhibits an A'BCDEFG topology (7 strands), although while sharing Ig-fold it exhibits a very different sequence pattern and may result from convergent evolution [63]. The A' strand in Cadherins corresponds to the A strand designation in the literature [66]. Classical cadherins can exhibit an A*/A strand split where its most N-terminal segment, called the "A* strand", provides an adhesive mechanism between cells by swapping between N-terminal (EC1) domains [67]. Both FN3 and Cadherins, however, show key Tyrosine residues conserved in strands C and F, where the Tyr in F strand might correspond to the Tyr corner of the F strand in the previous four canonical domains. (See supplement files S1 Data to S6 Data for multiple sequence alignments used in producing sequence logos in Fig 3).

from a structural and topological standpoint. We can refer to the diverse set of Ig-domain types as either *topo-structural* variants or *isotypes*.

Three taxonomy databases classify Ig-domain structural variants, beyond the common V/I/C1/C2-set nomenclature: **1) SCOP** [68,69] classifies the Immunoglobulin fold under its hierarchical nomenclature as immunoglobulin-like beta sandwich b.1 containing 33 superfamilies, where b.1.1, for example, contains the main 4 variants V-set, C1-set, C2-set, I-set. Beyond these domains others that vary in topology are classified under b.2 (Common fold of diphtheria toxin/transcription factors/cytochrome f) for 10 superfamilies, b.3 (Prealbumin) with 8 superfamilies, b.7 (C2-domain like) with an additional 5 superfamilies, for **a total of 56 Ig and Ig-like topo-structural variants currently delineated**. **2) CATH** [70,71] classifies Immunoglobulin beta sandwiches as **2.60.40** in their CATH hierarchical nomenclature as containing 330 superfamilies, with 45422 domains in the PDB database. **3) ECOD** [72,73] classifies Immunoglobulin beta sandwiches in their XHTF hierarchical nomenclature, which emphasizes distant evolutionary relationships, groups superfamilies in 42 "H" homologous categories under the "X" level immunoglobulin-like and an extra immunoglobulin related category containing 8 superfamilies such as Superoxide dismutase or Prealbumin that show some "T" topology variations.

An understanding of the **Ig-fold plasticity** leading to numerous tertiary and quaternary architectural complexes is required to decipher the myriad of molecular mechanisms underlying their biological functions, to design and develop therapies to correct dysfunctions, or to engineer new functions. In the following sections, we consider this body of experimental data on the Ig-fold to define a common, universal, residue numbering system. It will enable the parallel analysis of Ig-domain variants in terms of their diversity as well as the analysis of higher levels of complexity found in quaternary assemblies, from for example TCR/CD3 or BCR complexes and/or their co-receptors such as CD4, CD8 or CD79 at the surface of cells, but also from Ig-like domains such as Lamins and Arrestins.

## Ig, Ig-like and Ig-extended domains

All structurally-conserved folds and superfamilies exhibit sequence variations. Among them, some superfolds that are highly successful in evolution, such as 7-transmembrane G protein coupled receptors (GPCRs), kinases, Src homology 3 (SH3) domains, and Oligonucleotide/oligosaccharide-binding (OB) folds, can withstand very low sequence conservation to fold with an invariable framework topology, where structural insertions and deletions are localized in loops. **The Ig-fold is unique in that it can also vary its topology**. The topological variation of the Ig-fold is in itself a **paradox** that can be resolved by admitting that insertions can themselves form strands that extend a beta sandwich fold core, and that some strands around that core can swap between the two sheets of the beta sandwich. Variations in sequence, strand topology and loops offer an immense structurally and chemically diverse repertoire to the Ig-fold that enables it to engage in a spectrum of molecular interactions spanning multiple biological functions (both mechanical and biochemical).

The difficulty is in capturing the immense plasticity of the Ig-fold and to qualify, let alone quantify, the variability in sequence, topology, and structure. We first review the elements of plasticity in the Ig-fold and then define a universal residue numbering scheme that can be used across all topo-structural variants seen in nature.

## Structural invariance and topological variability of the Ig-fold

**The irreducible structural signature of the Ig-fold.** The plasticity observed across all topo-structural variants of the Ig-fold [49,51,52] is significant. However, they all exhibit a structural signature composed of four central beta-strands B-C and E-F using the classical Ig-domain nomenclature [46,48,74]. These 4-strands form the core central beta sandwich supersecondary structure (4s-SSS) as two straddling 2-beta strands motifs B-C and E-F intertwined in a pseudo-symmetric arrangement, and are supplemented by a 5th G strand to give a 5-strands supersecondary structure signature for the Ig-fold (5s-SSS) (see Fig 4). The two substructures B-C and E-F, straddle the two sheets of the sandwich through the BC loop (CDR1) and the EF loop barrel exhibiting a C2 pseudosymmetry [61]; the G strand at the

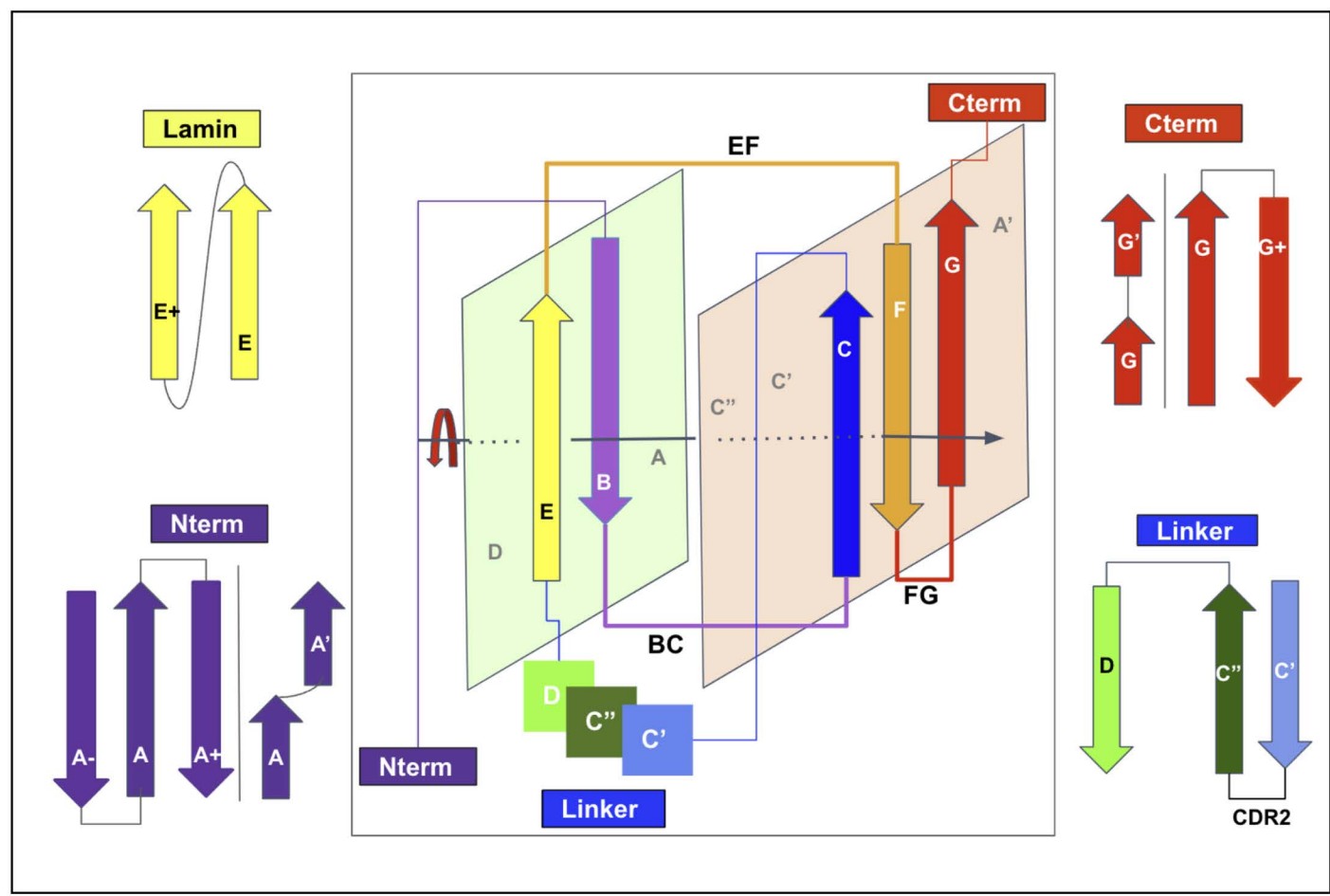

**Fig 4. Schematic representation of the Ig-fold structural signature consisting in strand B-C, E-F, and G strands arranged as displaced in the central panel, and main regions of lateral variability: NTerm (strand A), Cterm (strand G) and Linker (Strands between C and D), as well as an example of a less known lateral strand addition.** (See text).

C terminus forms the FG loop, known as CDR3 (complementary determining region 3) in antibodies' variable domains. Two to five lateral strands, comprising A/A' at the N terminus and C'/C"/D between C and E strands, define and differentiate the four classes of canonical Ig-domains (Figs 3 and 4): IgC1, IgC2, IgI and IgV [46,74].

Interestingly the central 4 strands (BCEF, 4s-SSS) represent a universal supersecondary structure motif common not just to the Ig-fold, but also to the jelly-roll fold as originally defined by Jane Richardson [75], suggesting that this central supersecondary structure may act as a stable folding nucleus for beta-sandwich folds to accommodate a myriad of topo-structural variants. Jelly rolls represent, after the Ig-fold, the second largest taxonomic group in structural classifications of SCOP, CATH, and ECOD, and these two groups represent the overwhelming majority of beta sandwich structures that in turn, as a class, represent 30% of all folds with beta secondary structures [76].

**Qualifying the variability of domains sharing the Ig-fold.** While the Ig-fold has been recognized in numerous topo-structural variants, detailed comparisons across all of them are difficult: in addition to variable sequence identity and variable lengths of the strands and loops as in any structural family, they exhibit different tilt angles between the two beta-sheets of the sandwich, and variable topologies with different number of strands in the well-known V/C1/C2/I sets. In some variants, additional strands may appear at the N-terminus to the A strand, interspaced structurally between the A and G strands, or at the C-terminus to the G strand, both on Sheet 1 (light green plane in Fig 4) and/or Sheet 2 (light orange plane in Fig 4), and in parallel or antiparallel with the adjacent strands. Variations within some strands can result in Ig variants with striking differences as some strands can form bulges, and/or can split with loop insertions, and split strands, or overall strands may swap between Sheet 1 and Sheet 2. This is a basis for the well-known A/A' strand split with its first half on Sheet 1 (ABED) and its second half (called A') on Sheet 2 (A'GFCC') in IgV and IgI domains. Some IgV domains can also swap the C" strand at the tertiary level between Sheet 1 and Sheet 2, as observed in crystal structures of TCR IgV or CTLA-4 IgV domains. That C" strand may also become disordered as in PD1, or be absent altogether as in PD-L2 or VNAR domains. In addition, beta bulges are common in beta strands A and G. Therefore, it is not surprising that when continuously numbering an Ig-domain, a specific structurally-conserved position, common to all members of the fold, gets a different residue number in different domains, even with simple variations. Considering the immense plasticity in strand topology, defining a universal residue numbering system becomes an important challenge.

## Towards positional structural bioinformatics

**Positional bioinformatics.** Since the very beginning, the field of bioinformatics has been focused on sequence alignment to compare the *relative* sequence identity between two or more protein sequences at given *positions*. However, even for highly similar sequences, the ***sequence numbers*** of corresponding (mapped) positions in alignments vary, due to insertions/deletions accumulated during evolution. A reference numbering system common to all members of a multiple sequence alignment (MSA), can enable *positional* comparisons in sequence, structure, and function. Since sequence datasets and especially structural datasets are becoming larger, the ability to compare similar positions of residues in sequence, topology, tertiary and quaternary structure becomes a necessity to capture position-based interaction patterns that underlie biological functions.

Examples of reference residue numbering systems in proteins are not legions for proteins: two main ones have been defined for domains with a fixed topology: for immunoglobulin variable domains (Kabat numbering, [77]) and for GPCRs (Ballesteros-Weinstein numbering,

[78]). In the realm of RNA, studies on the ribosome benefit from a reference numbering, using E.Coli as a reference species to enable comparisons and delineating evolution across species [79–81]. But the most impressive use of a reference numbering is in the field of genomics to annotate DNA base pair positions [82]. Genome reference numbering highlights a transition from comparative to positional bioinformatics. In fact, one may wonder where genomics would be today if there was no common reference numbering for the community to share annotations on the human genome, and how difficult it would be to discuss mutations, transcription factor binding sites, ORFs, introns and exons positions in genes, or any other positional feature. One can argue that a standardized nomenclature and a reference numbering system allows the scientific and medical community to annotate, analyze, compare, and coalesce information in a consistent manner on a biomolecular family of importance. Establishing a reference residue numbering scheme for proteins that share the Ig-fold is needed despite its inherent difficulty in representing the many topological and structural variations, and this is exactly what we seek: a universal numbering scheme that enables comparisons in structurally-conserved residue positions across all topological variants of Ig-domains.

**Requirements for a numbering scheme describing the Ig-fold.** Ig-domain families offer an elaborate diversity in structure, topology, sequence and function. Focusing on Ig-domains and seeking a universal residue numbering scheme adaptable to all Ig-domain topological variants, we aim at enabling parallel analyses of **tertiary domains, chains**, and their quaternary **assemblies**. Ig domains form a myriad of Ig-Ig complexes with diverse geometries that can, in some cases, be considered as "**quaternary topologies**" and even "**quaternary folds**", as for example in antibody Fab, IgV and IgC1 dimers of heavy and light chains. While we expect to learn evolutionary relationships from tertiary domains and chains made of Ig-domains in tandem, we should also expect to learn about co-evolutionary relationships between receptors and ligands made of and interacting with Ig-domains, given their remarkable property of self-assembly as in antibodies heavy and light chains, or in cell surface receptors-ligand associations in either cis or trans orientations. Ig-domains assemble in tandem in chains and in quaternary complexes, which brings three levels of complexity (see below). From a domain level reference numbering scheme one can expect the ability to then compare Ig domain quaternary complexes.

A universal numbering scheme common to all Ig, Ig-like and Ig-extended domains is intended to enable direct comparisons between domains and their associations across very large datasets and opens the door to proteome scale analysis. While a numbering scheme may be used for machine learning, we seek one that is human readable, following examples such as Kabat numbering [77] on IgVH and IgVL domains of antibodies or Ballesteros-Weinstein numbering [78] in GPCRs (see later).

**Topo-structural variants reference dataset.** The structural diversity of Ig-domains is captured by classifications such as ECOD, CATH and SCOP described earlier. These classifications rely on overall pattern matching between domains, using structural alignments. They offer distinct variant classes in structure and topology, albeit in slightly different ways. The drawbacks of such detailed classifications are that i) they separate topo-structural variants in silos and ii) they do not offer a way to compare and relate topologies. This makes the study of tertiary plasticity across them difficult. To establish a universal numbering scheme for the Ig-fold, we first analyzed a dataset of Ig-domain superfamilies, classified as b.1 in SCOP, and containing domains from plant, fungi, archaea, bacteria, and viruses to capture and evaluate structural diversity across life. We then selected representatives of cytoplasmic proteins, nuclear proteins, and cell surface receptors in the immune, nervous, vascular and muscular systems from the animal kingdom favoring a human origin. Table 1 lists a selected reference set of Ig-domains that will be used as "templates". While biased towards the human species,

**Table 1. Reference set of diverse topological and structural variants of Ig domains** from the SCOP b.1 Superfamily. (see supplement file S7 Data, for the corresponding spreadsheet). It includes surface receptors in the immune, nervous, and vascular systems, cytoplasmic proteins and enzymes, transcription factors and nuclear proteins.

| Filename | Structural Classification | Protein | Template | SCOPe | PDB ID | Chain ID | RefSeq |
|---|---|---|---|---|---|---|---|
| B2Microglobulin_7phrL_human_C1.pdb | C1 | **B2-Microglobulin** | Ig | b.1.1.2 | 7phr | L | P61769 |
| BTLA_2aw2A_human_Iset.pdb | I-set | **BTLA** | Ig | b.1.1.4 | 2aw2 | A | Q7Z6A9 |
| CD19_6al5A_human_C2orV-n1.pdb | CD19 | **CD19** | Ig | Unk. | 6al5 | A | P15391 |
| CD2_1hnfA_human_C2-n2.pdb | C2 | **CD2** | Ig | b.1.1.3 | 1hnf | A | P06729 |
| CD2_1hnfA_human_V-n1.pdb | V | **CD2** | Ig | b.1.1.1 | 1hnf | A | P06729 |
| CD28_1yjdC_human_V.pdb | V | **CD28** | Ig | b.1.1.1 | 1yjd | C | P10747 |
| CD3d_6jxrd_human_C1.pdb | C1 | **CD3d** | Ig | b.1.1.4 | 6jxr | d | P04234 |
| CD3e_6jxrf_human_C1.pdb | C1 | **CD3e** | Ig | b.1.1.4 | 6jxr | f | P07766 |
| CD3g_6jxrg_human_C2.pdb | C2 | **CD3g** | Ig | b.1.1.4 | 6jxr | g | P09693 |
| CD8a_1 cd8A_human_V.pdb | V | **CD8a** | Ig | b.1.1.1 | 2atp | A | P01731 |
| Contactin1_3s97C_human_Iset-n2.pdb | I-set | **Contactin1** | Ig | b.1.1.0 | 3s97 | C | Q12860 |
| FAB-HEAVY_5esv_C1-n2.pdb | C1 | **FAB-CH1** | Ig | b.1.1.2 | 5esv | C | 5ESV_C |
| FAB-HEAVY_5esv_V-n1.pdb | V | **FAB-VH** | Ig | b.1.1.1 | 5esv | C | 5ESV_C |
| FAB-LIGHT_5esv_C1-n2.pdb | C1 | **FAB-CL1** | Ig | b.1.1.2 | 5esv | D | 5ESV_D |
| FAB-LIGHT_5esv_V-n1.pdb | V | **FAB-VL** | Ig | b.1.1.1 | 5esv | D | 5ESV_D |
| ICOS_6x4gA_human_V.pdb | V | **CD278 (ICOS)** | Ig | b.1.1.1 | 6x4g | A | Q9Y6W8 |
| JAM1_1nbqA_human_Iset-n2.pdb | I-set | **JAM1** | Ig | b.1.1.4 | 1nbq | A | Q9Y624 |
| LAG3_7tzgD_human_C2-n2.pdb | C1 | **LAG3** | Ig | Unk. | 7tzg | D | P18627 |
| LAG3_7tzgD_human_V-n1.pdb | V | **LAG3** | Ig | Unk. | 7tzg | D | P18627 |
| MHCIa_7phrH_human_C1.pdb | C1 | **MHCI alpha** | Ig | b.1.1.2 | 7phr | H | P04439 |
| Palladin_2dm3A_human_Iset-n1.pdb | I-set | **Palladin** | Ig | b.1.1.4 | 2dm3 | A | Q8WX93 |
| PD1_4zqkB_human_V.pdb | V | **CD279 (PD1)** | Ig | b.1.1.1 | 4zqk | B | Q15116 |
| PDL1_4z18B_human_V-n1.pdb | V | **CD274 (PDL1)** | Ig | b.1.1.1 | 4z18 | B | Q9NZQ7 |
| Siglec3_5j0bB_human_C1-n2.pdb | C1 | **Siglec-3 CD33** | Ig | Unk. | 5j0b | B | P20138 |
| Titin_4uowM_human_Iset-n152.pdb | I-set | **Titin** | Ig | Unk. | 4uow | M | Q5VST9 |
| VNAR_1t6vN_shark_V.pdb | V | **VNAR** | Ig | b.1.1.1 | 1t6v | N | 1T6V_N |
| C3_2qkiD_human_n1.pdb | FN3-like | **C3 Complement** | Ig-like | b.1.29.2 | 2qki | D | P01024 |
| Contactin1_2ee2A_human_FN3-n9.pdb | FN3 | **Contactin1** | Ig-like | b.1.2.0 | 2ee2 | A | Q12860 |
| ECadherin_4zt1A_human_n2.pdb | Cadherin | **E-Cadherin** | Ig-like | b.1.6.1 | 4zt1 | A | P12830 |
| GHR_1axiB_human_FN3-n1.pdb | FN3 | **GHR** | Ig-like | b.1.2.1 | 1axi | B | P10912 |
| IL6Rb_1bquB_human_FN3-n2.pdb | FN3 | **CD130 (IL6R-beta)** | Ig-like | b.1.2.1 | 1bqu | B | P40189 |
| IL6Rb_1bquB_human_FN3-n3.pdb | FN3 | **CD130 (IL6R-beta)** | Ig-like | b.1.2.1 | 1bqu | B | P40189 |
| InsulinR_8guyE_human_FN3-n1.pdb | FN3 | **Insulin Receptor** | Ig-like | b.1.2.1 | 8guy | E | P06213 |
| InsulinR_8guyE_human_FN3-n2.pdb | FN3 | **Insulin Receptor** | Ig-like | b.1.2.1 | 8guy | E | P06213 |
| NaCaExchanger_2fwuA_dog_n2.pdb | FN3-like | **NaCa Exchanger** | Ig-like | b.1.27.1 | 2fwu | A | P23685 |
| NaKATPaseTransporterBeta_2zxeB_spurdogshark.pdb | Unknown | **NaK ATPase Transporter-Beta** | Ig-like | b.1.32.1 | 2zxe | B | P05027 |
| ORF7a_1xakA_virus.pdb | Unknown | **ORF7a** | Ig-like | b.1.24.1 | 1xak | A | P59635 |
| RBPJ_6py8C_human_Unk-n1.pdb | Unknown | **RBPJ** | Ig-like | Unk. | 6py8 | C | Q06330 |
| RBPJ_6py8C_human_Unk-n2.pdb | Unknown | **RBPJ** | Ig-like | Unk. | 6py8 | C | Q06330 |
| Sidekick2_1wf5A_human_FN3-n7.pdb | FN3 | **Sidekick-2** | Ig-like | b.1.2.1 | 1wf5 | A | Q58EX2 |
| TP47_1o75A_bacteria.pdb | Unknown | **Tp47 Lipoprotein** | Ig-like | b.1.20.1 | 1o75 | A | P29723 |
| ASF1A_2iijA_human.pdb | FN3-like | **ASF-1A Histone Chaperone** | Ig-extended | b.1.22.1 | 2iij | A | Q9Y294 |
| BArrestin1_4jqiA_rat_n1.pdb | FN3-like | **B-Arrestin-1** | Ig-extended | b.1.18.11 | 4jqi | A | P29066 |
| CoAtomerGamma1_1r4xA_human.pdb | Unknown | **CoAtomer-Gamma-1** | Ig-extended | b.1.10.3 | 1r4x | A | Q9Y678 |

*(Continued)*

**Table 1.** (Continued)

| Filename | Structural Classification | Protein | Template | SCOPe | PDB ID | Chain ID | RefSeq |
|---|---|---|---|---|---|---|---|
| CuZnSuperoxideDismutase_1hl5C_human.pdb | FN3-like | **CuZn Superoxidase Dismutase** | Ig-extended | b.1.8.2 | 1hl5 | C | P00441 |
| Endo-14-BetaXylanase10A_1i8aA_bacteria_n4.pdb | Unknown | **Endo-1,4-Beta--Xylanase-A** | Ig-extended | b.1.9.2 | 1i8a | A | Q60037 |
| IsdA_2iteA_bacteria.pdb | Unknown | **IsdA NEAT domain** | Ig-extended | b.1.28.1 | 2ite | A | Q2FZE9 |
| LaminAC_1ifrA_human.pdb | Unknown | **Lamin-A/C** | Ig-extended | b.1.16.1 | 1ifr | A | P02545 |
| MPT63_1lmiA_bacteria.pdb | FN3-like | **MPT63** | Ig-extended | b.1.19.1 | 1lmi | A | P9WIP1 |
| TEAD1_3kysC_human.pdb | FN3-like | **TEAD1 (TEF1)** | Ig-extended | b.1.18.26 | 3kys | C | P28347 |
| TP34_2o6cA_bacteria.pdb | Unknown | **TP34** | Ig-extended | b.1.33.1 | 2o6c | A | P19478 |
| VISTA_6oilA_human_V.pdb | V | **VISTA (VSIR)** | Ig-extended | b.1.1.1 | 6oil | A | Q9H7M9 |

the selected Ig-domain representatives are structurally fit for the identification and labeling of Ig-domains in all domains of life. In fact, while the common numbering scheme defined on this b.1 dataset (Table 1) was initially aimed at covering Ig- and Ig-like domains, we found that it could be easily adapted to cover Ig-extended domains, i.e., topo-structural variants classified in various fold categories such as SCOP b.2 (like p53), b.3 (like prealbumin), b.7 (like C2-domain) by allowing topological insertions of strands, instead of introducing more templates covering these variants, although the need for high precision may require more templates in the future, and other algorithmic improvements.

**The original Kabat numbering and its variants for IgV domains.** The idea of a domain-level positional analysis, originally in sequence, is due to Elvin A. Kabat in late 1960s [77,83–85] as he started compiling and tabulating a large number of antibody sequences to form the Kabat database [86], *observing that* **positions in antibody sequences** *could be associated with invariant vs. highly variable residue types in certain segments of their sequence*, named Framework (FR) vs. hypervariable regions (CDR: Complementary Determining Regions) [85]. Kabat's delineation of functional subdomains (Framework) and hypervariable regions (CDRs) responsible for binding antigens, was later confirmed when the first Bence-Jones protein (an $IgV_L$ dimer) [65,84] and antibody structures were determined [50,53,87]. This was one of the first pieces of evidence of the close relation between sequence, structure and function of antibody proteins. The "Kabat numbering" scheme was assigning positional numbers in sequence common to all antibody IgV domain light chains ($IgV_L$) and heavy chains ($IgV_H$) (independently), i.e., each residue in the antibody domain is assigned a unique number based on its position in sequence, allowing an easy reference and comparison on specific locations within antibody $IgV_L$ and $IgV_H$ domains.

As more sequences and structures accumulated some shortcomings of the initial Kabat numbering appeared and a number of improvements to the numbering scheme were proposed: the Chothia numbering [88], and the improved Chothia, also named Martin numbering [89] to improve Kabat's numbers based on structure. Another limitation of the Kabat numbering is the use of different position numbers in VH and VL domains. Improvements and extensions were later proposed to describe antibody and TCR Ig domains with the IMGT numbering scheme [90–92]. Variations on the latter have also been proposed as yet another numbering scheme (AHo) [93]. While Kabat numbering has been used for a long time, only recently were tools developed to map Kabat and other numbers to antibody variable domains [94].

In the following sections, we present a universal Ig residue numbering scheme applicable to Ig, Ig-like, and Ig-extended domains, followed by a demonstration of its strength in comparative analysis of Ig-Ig interactions in antibodies, quaternary interactions beyond antibodies, and complex Ig-chain interfaces.

## Results and discussion

### A universal numbering scheme

In this work, a new univernal Ig residue numbering scheme (called IgStrand) is presented, which enables positional bioinformatics driven functional analyses of structurally conserved residue positions in all Ig and Ig-like folds in nature's proteome. The numbering scheme is described using the classical Ig-folds (IgV, IgC1, IgC2, IgI) and then the flexibility built into the scheme is highlighted through its application to Ig-extended domains that have additional strands compared to the classical Ig-folds. The limits of the numbering scheme are further tested through its application to jelly-rolls that appear to be similar to Ig-folds when considering additional strand plasticity. The true value of the numbering scheme is then presented through its application to several examples that highlight the structural and functional diversity of Ig-domains, higher complexity tertiary constructs (Ig-chains), and quaternary complexes (Ig-assemblies) that involve Ig-Ig domain interfaces.

### Definition of the IgStrand numbering

The IgStrand numbering scheme (IgStRAnD acronym for Ig Strand Residue Anchor Dependent) is inspired by Kabat's original numbering [77] for IgVH and IgVL domains, as well as the Ballesteros-Weinstein (BW) GPCR numbering scheme [78]. The latter uses numbers centered on secondary structure elements (SSE) and their most conserved residues, considered as sequence "**anchors**", assigning them a number 50, then counting both positively and negatively in each SSE, where each SSE is assigned a number i=1,2,...7 for 7-helical membrane proteins. For example, 2.50 would be "helix 2, number 50" for the most conserved residue in TM2, an aspartate, in class A GPCRs. The residue that appears before this conserved aspartate in sequence would be numbered 2.49 and the one that appears after it would be numbered 2.51, and so on. The numbers can be considered hierarchical, but also simply decimal numbers (i.e., 2.50 or 250 in this case).

Similarly, the IgStrand numbering scheme defines anchors with a number 50, but these are defined structurally and are not based solely on sequence conservation to account for the immense variability in sequence in topo-structural variants. Besides, this scheme uses 4 digits to number residues in an Ig domain (ijxx). The numbers are hierarchical but also decimal to ensure sequence continuity. The first digit "i" (1000th place) defines the strand number ranging from 1 to 9 (i.e., 1jxx to 9jxx) for the nine main canonical strands (A, B, C, C', C", D, E, F, G) present in the most common Ig topologies such as the Immunoglobulin variable domain. The second digit "j" (100th place) is set to "5" for each of the main canonical strands (A, B, C, C', C", D, E, F, G). This digit can deviate from "5" for strand insertions between canonical strands. For example, if an Ig domain has an extra strand after G strand (like in CD3$\gamma$), that strand may be named G+ and its "j" value would then be "6", so its residues will be numbered 96xx. The third and fourth digits assign residue numbers centered around "50" inspired by the BW scheme used in GPCRs [78]. The structurally conserved anchor residue positions in those strands are numbered 1550, 2550, 3550, 4550, 5550, 6550, 7550, 8550, and 9550. The other residues in those strands are numbered relative to the anchor residues based on the protein sequence, e.g., the residue that appears right before A strand's anchor residue is numbered 1549 and the residue that appears right after that anchor residue is numbered 1551. A subtle difference between IgStrand anchor residues (xx50) and BW anchor residues (x.50) is that due to the beta-strand topology of Ig-folds the IgStrand anchor residues form backbone hydrogen bonds and hence may not be conserved, whereas, BW anchor residues in GPCRs are highly conserved. The Ig strands typically have between 5–15 residues in each strand, so the last two digits will never reach a number below "00" or above "99". In many Ig-domains, the A strand

splits into A/A' strands, such that A continues to sheet up with the B strand but A' sheets up with the G strand. The anchor residue on A' strand is numbered 1850, to account for additional strands between A and A', e.g., A+ strand seen in p53 (16xx) and a potential possibility of A++ strand (17xx).

**Defining positional structural anchors "ij50" in the Ig-fold beta strands.** To define structurally conserved anchor positions, a reference set of Ig domains providing topological diversity were structurally aligned to the IgVH domain, which has been extensively studied. This alignment (Fig 3) confirmed known conserved residues and uncovered the conservation of specific residue positions within each strand and their network of interactions defining the interior of the Ig-folds (see Methods). The most structurally conserved position in each strand was selected as the residue anchor for its respective strand. All anchors selected are in-facing residues as they form an interconnected network of backbone and sidechain interactions, crucial to the folding and stability of the Ig-fold. Anchors chosen structurally coincide with key conserved residues in sequence for well known superfamilies such as Immunoglobulins (SCOP b.1.1.1), but are otherwise highly variable across all topo-structural variants (Fig 3). Figs 5 and 6 highlight highly conserved residues in variable and constant domains of antibodies, TCRs and other immune receptors. The specific residue position chosen as anchor in each of the classical 7–9 strands (A, B, C, C', C", D, E, F, G), as well as in the characteristic A strand split (see later), forming a so-called A' strand in IgV and IgI domains, is given a number "50" regardless of the Ig variant type, the lengths of the strands, the loops, angle between the two beta-sheets, or sequence variability.

**Defining canonical and additional beta strand numbers (i and j).** The classical 7–9 strands for canonical Ig domains A, B, C, C', C", D, E, F, G are given the number i=1,2,3,4,5,6,7,8,9 respectively (1000's) (see Fig 5A). The second digit is fixed to j=5. To allow for strand insertions, the second digit j can be numbered negatively or positively from it, the special case of A' strand getting j=8 (see Fig 5A). The third and fourth digits correspond to the residue number in a given strand and beyond, counting both positively and negatively from the anchor residue number 50, as in BW numbering for GPCRs, so a strand of 7 residues for example could run from ij47 to ij53 around the anchor ij50. The numbers will then run further than the strand in both directions to accommodate strand length variability as well as loop numbering from and to the next strand. In this scheme, the N-terminal half of the loops inherit the continuous numbering forward from the previous strand and the C-terminal half of the loops inherit the continuous numbering backward from the following strand (see Methods). The N and C term residues, before the first and the last strand respectively, inherit the numbering going backward and forward respectively from those strands. So, all residues in the Ig domain are accounted for in this numbering scheme. This numbering does not currently explicitly consider the sheet in which a strand lies, as in some alternative and hierarchical numbering schemes that we have considered. The sheet information would be a useful future extension to the IgStrand numbering scheme.

Anchor selection started with the disulfide bridge cysteines on the B strand (igs# 2550) and the F strand (igs# 8550), with respect to which the rest of the anchors were subsequently selected. The GFCC' anchors form an extended stretch of backbone-backbone interactions, which includes the conserved tryptophan on the C strand (igs# 3550), while C" anchor is shifted negatively by one residue with respect to interstrand backbone interactions on the GFCC'C" sheet. The A strand backbone neighbor of the B strand cysteine was selected as the A strand anchor (igs# 1550), which is often a glutamate or glutamine that interacts with the backbone of the F strand cysteine (igs# 8550) in variable domains. Due to the rotation of the two beta-sheets in the Ig domain beta-sandwich, the anchors are shifted by two residue positions negatively for the D strand (igs# 6550) and positively for the E strand (igs# 7550).

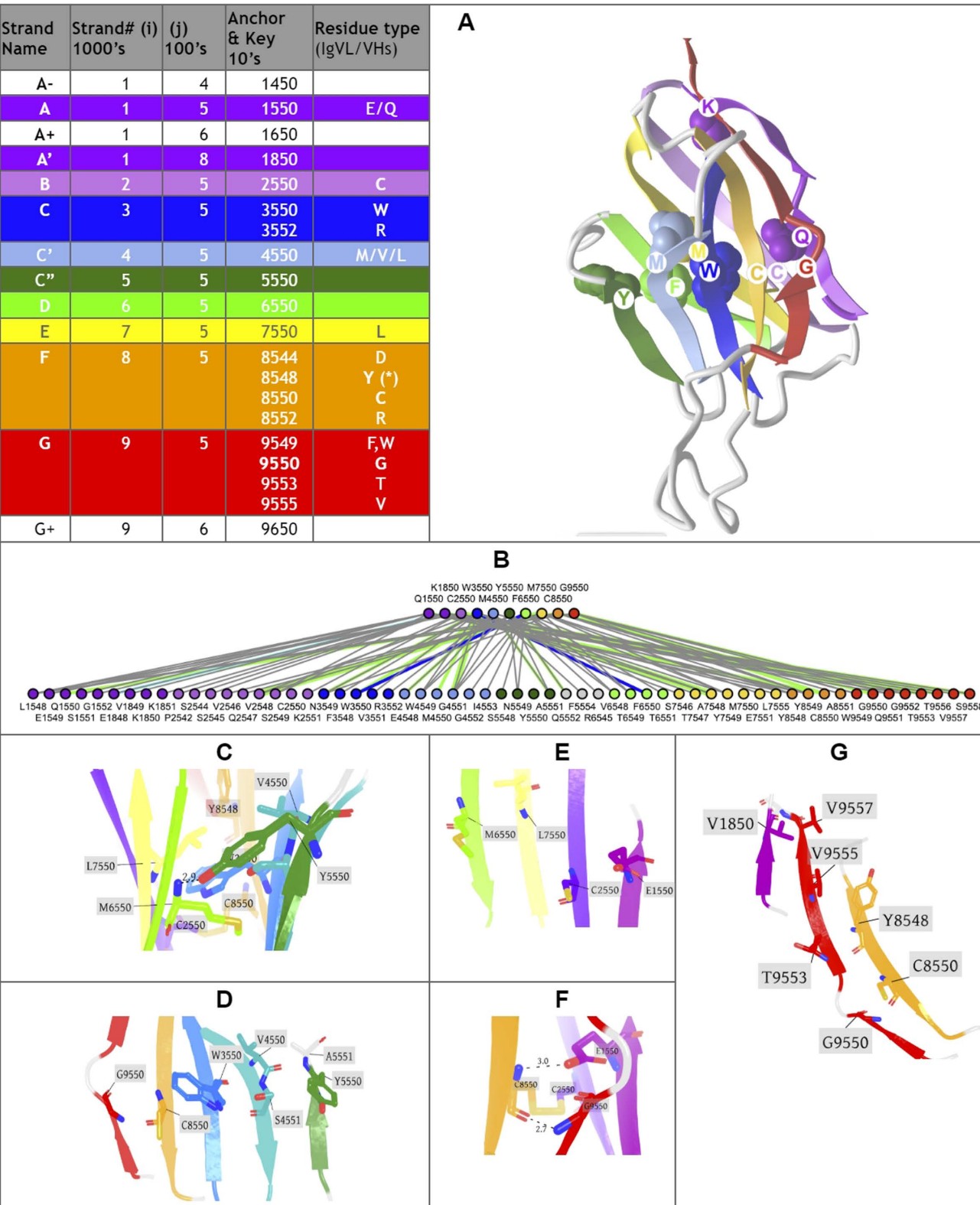

**Fig 5. Anchor positions. A) IgStrand numbers for anchors and additional key residues** in important IgV domains Definitions for Classical Strands (1000's), Non-classical Strands (100's) with Anchor numbers (50s). Residue types are shown for well-known IgV examples as in antibodies where sheet 1 is composed of strands ABED and sheet 2 (A')GFCC'(C"). The A strand anchor residue is given the number 1550, where it is the first strand (A) on Sheet 1. A' strand anchor is given the number 1850 it is the second half of the A strand swaps to Sheet 2 adjacent to the G strand in parallel. The B

strand anchor is 2550. C strand anchor is 3550 and is on Sheet 2. C' strand anchor is 4550 and is on Sheet 2. C" strand anchor is 5550 on Sheet 2, but it can swap to Sheet 1. D strand anchor is 6550. The E strand anchor is 7550. The F strand anchor is 8550. The G strand anchor is 9550. Additional, non-classical strands can also appear in Ig-extended domains. This is taken into account by the second digit, for example the anchor of the strand appearing before the A strand (A- strand) will have the IgStrand number 1450, while the anchor of a strand A+ after the A strand will be 1650, and for a G+ strand appearing after a G strand similarly 9650. The anchors form a network. **B)** shows the extensive residue interaction network between the anchors core and residues in all strands of the fold. **C)** shows sidechain and backbone representations of all anchors with respect to the overall fold. **D)** shows the positions and backbone connections of the anchors in the GFCC'C" Sheet 2. **E)** shows **D** 6550 and **E** 7550 strand anchors vs. A 1550 and **B** 2550 anchors on the ABED Sheet 1. **F)** shows the interaction of A strand anchor 1550 with the backbone of 8550, forming the signature bulge in the G strand of IgVs. **G)** shows the position of 1850 in the A' strand with respect to the F and G strand residues. (PDBid used: 5ESV_H for heavy chain variable domain).

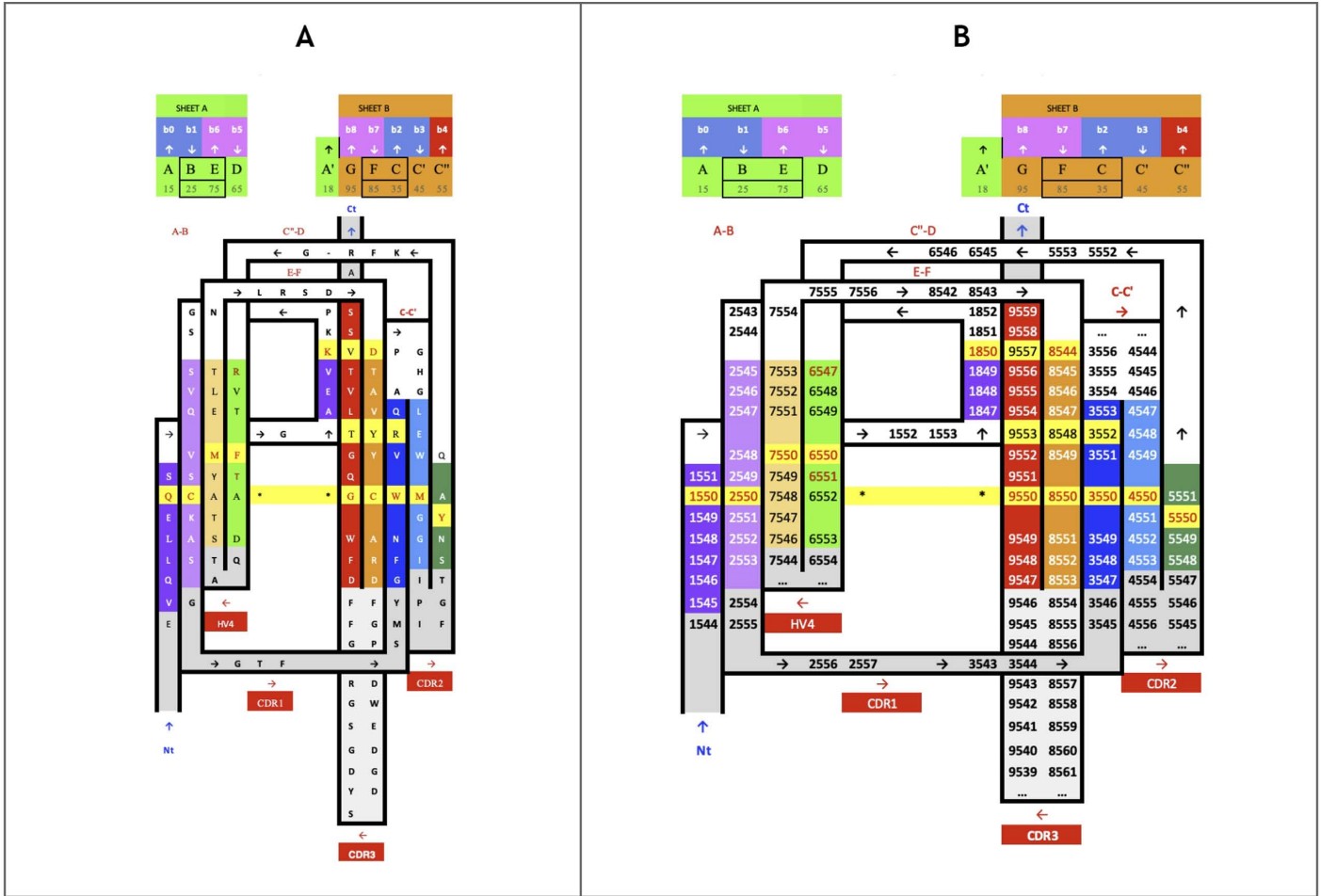

**Fig 6. IgV domain Proteomap A) Sequence/Topology Map (PDBid 1RHH)** of the VH domain show both the the sequence and topology simultaneously https://structure.ncbi.nlm.nih.gov/icn3d/share.html?DTtCtLWevzwS2XkJ8&t=1RHH. B) IgStrand numbers corresponding to IgV domains with the AA'BCC'C"DEFG topology.

Particularly, this igs# 7550 is highly conserved as leucine and stabilizes the C strand tryptophan (igs# 3550) and may play a critical role in folding. In a special case, since the A' is an extension of the A strand that is located at the edges of the Ig domain beta-sandwich, the most C-terminal inward-facing residue was selected as the A' anchor position (igs# 1850), which predominantly provides hydrophobic contacts that stabilize the EF loop and the core of the Ig domain.

**IgStrand Structural Anchors.** The structural anchors provide a network of backbone interactions along the two sheets of the beta sandwich as well as non-bonded core stabilizing hydrophobic interactions, and in the case of canonical domains the formation of covalent cysteine bridges.

**The A strand** anchor position (igs# 1550) (see Fig 5A: i=1,j=5) is typically occupied by residues E/Q in IgVH, IgVL, and many other IgV domains, while in others it may be a hydrophobic residue. E/Q at position igs# 1550 provides a polar sidechain interaction with the backbone of F strand Cysteine igs# 8550, stabilizing the classical BF disulfide bond. **The A strand** represents a structurally variable region, even within canonical Ig domains. **The A' strand**, as it is named conventionally, may be pictured as an A strand that splits, with the second half (A') swapping from Sheet 1 (ABED) to Sheet 2 (A'GFCC'C") in IgV or IgI domains. This gives that strand an additional A' anchor (igs# 1850) on the edge of the strand before the A'B loop, indirectly connected to the hydrophobic core of the Ig domain through a network of interactions. The split A/A' strand provides a strong connection between the two sheets of the sandwich at the N-terminus. The A' strand anchor residue (igs# 1850) is in-facing, and the prior in-facing residue in the strand (igs# 1848) forms a parallel backbone H-bond with the residue (igs# 9554) on the G strand, itself two residues downstream of the G strand residue (igs# 9552) that is adjacent to and forms a backbone pair with the residue (igs# 8548) on the F strand, which is two residues upstream of the F strand anchor (8550) (Figs 5G, 6, and B in S1 Text). We opted to assign the A' strand an anchor position because many Ig domains lack the A strand but contain only an A' strand, such as CD4, CD19, and many others. This A' anchor position is structurally important for closing the ends of the two beta-sheets by providing a hydrophobic contact and rendering the core as solvent-inaccessible.

As we have seen, the core structural signature of an Ig-fold is composed of the four central beta-strands **B,C,E,F**, forming two loops straddling the two sheets of the sandwich barrel: BC loop and EF loop. The corresponding anchors igs# 2550, 3550, 8550 and to a lesser extent 7550 in strands B, C, F, and E respectively (see Fig 5A) are highly conserved positions occupied by the residues CCW(L) [61], respectively. The highly conserved Cysteine anchors igs# 2550 and igs# 8550 form a disulfide bond in most canonical Ig domains (Figs 5C and 6), apart from some exceptions such as CD2. Another highly conserved residue is Trp on the C strand (igs# 3550) flanking the Cysteine bridge, which plays an important role in stabilizing the core of canonical Ig-folds. This Trp can also form hydrogen bond interactions through its indole ring amine group with residues on the E strand either directly with polar side chains or indirectly through a water molecule with backbone atoms (PDB ID: 5DK3).

In domains with a **C' strand** (IgV, IgI and IgC2) the anchor position (igs# 4550) is adjacent to the anchor on the C strand (igs# 3550) and is typically occupied by a hydrophobic residue such as Leucine. In most IgVs, a beta bulge on C' strand provides two consecutive in-facing residues due to trans backbone configuration of igs# 4550. Both in-facing residues form H-bond backbone interactions with the igs# 3550 backbone atoms on the C strand. Igs# 3550 also accepts a hydrogen bond from the backbone of igs# 5551 on the C" strand. The first of these two residues is chosen as the anchor since some Ig domains such as VNARs and IgIs have a very short C' strand that ends at igs# 4550 precisely before forming the C'D loop. IgC2 variants, which classically have a relatively long C' strand and no C", lack the two in-facing consecutive residues signature seen in most IgVs, and instead only have one in-facing residue that donates and accepts backbone hydrogen bonds with igs# 3550. Igs# 4550 provides additional stability to the hydrophobic core of the Ig-fold by forming vdW interactions with igs# 3550 and igs# 7550.

Igs# 5550, present predominantly in IgVs, lies one residue upstream from igs# 5551 that forms backbone hydrogen bond interaction with igs# 4550. The reason for this is that as

additional strands extend away from the BCEF core, the residues on these strands that continue the backbone interactions from core anchors shift away from the hydrophobic core of the Ig-fold due to the angle between the two beta-sheets. Oftentimes, these additional strands break off early and form short strands, as seen with C". Based on this, we selected the anchor of the C" to be an in-facing residue that interacts through its sidechain with core anchors, and this position appears to be conserved within subclasses of Ig-domains. Igs# 5550 is typically a Tyr in VH domains whose sidechain hydroxyl group interacts with the backbone of igs# 6550 on the D strand, providing a bulky side chain that protects the hydrophobic core as well as a polar interaction that stabilizes the intersheet distance (PDB: 5ESV). In VL domains, Arg appears to be highly conserved at the igs# 5550 position.

Igs# 6550 and 7550 are adjacent to one another and face each other through the alpha carbons of their backbones. Igs# 6550 is less conserved at the sequence level and the residue here can vary depending on the overall structure of the Ig-fold and the type of in-facing residues on the C' and C" strands, whereas igs# 7550 is highly conserved and is typically a Leucine in classical Ig-folds that stabilizes the Tryptophan igs# 3550 and may potentially be important for the folding of an Ig-domain. Both igs# 6550 and 7550 are shifted two positions away from the backbone network of interactions spanning from igs# 2550 on the B strand. The reason for this positional shift is similar to that of the C" (core interactions) strand.

## Ig-extended domains: Ig-fold topo-structural variable regions

Ig-fold canonical variants of **Ig- and Ig-like domains** vary in the following regions: **The linker region between C and E strands** differentiates the canonical immunoglobulin domain variants IgC1, IgC2, IgI, and IgV; **the N-terminus A strand** offers significant variability, with a continuous A strand in IgC1 domains, a split A/A' strand in IgI, IgV, and some IgC2 domains, or in some IgVs such as CD4 only a single A' strand is found; **the C-terminus invariant G-strand** often offers a strand split in the middle but remains on the same GFC(C') sheet in Ig-domains to form an FG loop called CDR3 in antibodies. This leads to a 5-strand supersecondary structure (5s-SSS) forming the irreducible Ig-fold structural signature (Fig 4).

Besides canonical Ig-domains topologies, **Ig-extended** domains exhibit more variations: **N-terminus strand additions,** beyond the classical A/A' strand split, can present additional strands that we name A+ or A- that can be positioned after or before in sequence and in structure on either sheet; and **C-terminus strand additions** that we name G+, G++, etc., can extend the Ig-fold on either sheet of the beta sandwich. **Strand insertions** also occur in a variety of positions in the two sheets of the beta sandwich. As mentioned above, insertions can lead to additional strands. An exhaustive survey is beyond the scope of the current paper, but we will see a number of examples in the following sections. The most surprising insertions leading to extended sheets may be stemmed in the central BC or EF loops leading to multiple variants: **insertion of a sequence between E and F** can lead to the formation of an additional E+ strand. It is characteristic of Lamins, extending the ABE sheet as ABEE+ similarly to the ABED sheet in canonical Ig-domains IgC1/I/V where the E+ strand is positioned as a D strand in Ig-domains but in reverse strand directions (see **Figs 2** and **4**). This introduces an asymmetry with the formation of an E+F straddling loop instead of a canonical EF loop in the 4s-SSS; similarly, **insertion of a sequence between B and C** can form, as in orthopoxviruses immune evasion proteins for example, a beta hairpin extending the GFC sheet in the same positions as the canonical C" and C' in IgV domains in reverse order. It introduces an asymmetry vs. the canonical BC straddling loop in the 4s-SSS and what may look like a permutation between strands C' and D. This type of Ig-domain extension is not shown in **Fig 2**, but can be seen later in the example of the monkeypox virus M2 protein in. A number of Ig-extended examples are presented hereafter.

Finally, **strand permutations** can also bring an additional level of combinatorial plasticity to the Ig-fold and help relate other sandwich folds to the Ig-fold itself. Circular permutations in sequence leading to structurally homologous structures are common in proteins. In C2 domains for example the A strand at the N terminus can be circularly permuted at the C terminus ([Fig 7]). Apparent simultaneous insertions/deletions as in the case of the Transthyretin/Prealbumin fold (SCOP b.7) of strands can also be seen as a pseudo-circular permutation where a G+G++ at the C terminus can be seen as replacing a possible A-A harpin at the N-terminus ([Fig 7]).

In summary, numerous domain variants can be classified as Ig-extended when considering strand insertions, permutations and N/C extensions, in beta sandwich sheets, but also deletions except for the four central B,C,E,F strands (4s-SSS). This can occur with N-terminal and C-terminal strands. The Ig-fold extensions are reviewed below through a number of examples found in nuclear, cytoplasmic and extracellular regions of cells in a diverse set of biological functions.

**Arrestins.**   Arrestin is particularly interesting as it is composed of two extended Ig-domains in tandem with an unusual head-to-head arrangement, as compared to the classical tail to head arrangement found in Ig-chains (in the N to C direction of Ig chains). The Arrestin domain [95] is a good example of an Ig-fold variant with an A- strand extension at the N-term that precedes that A strand, as well as an additional E+ strand following the E strand to form an A-ABEE+ sheet facing a GFCC' sheet. The tandem Arrestin results from the G strand of the Arrestin_N domain pipelined to the A- strand of the Arrestin_C domain. This A- strand of the C-terminus domain acts like an inversion strand for the C-terminal Ig domain and results in a head-to-head pseudo-symmetric structure ([Fig 8A]). **This shows the profound change in chain architecture obtained through the Ig domain A- strand extension**.

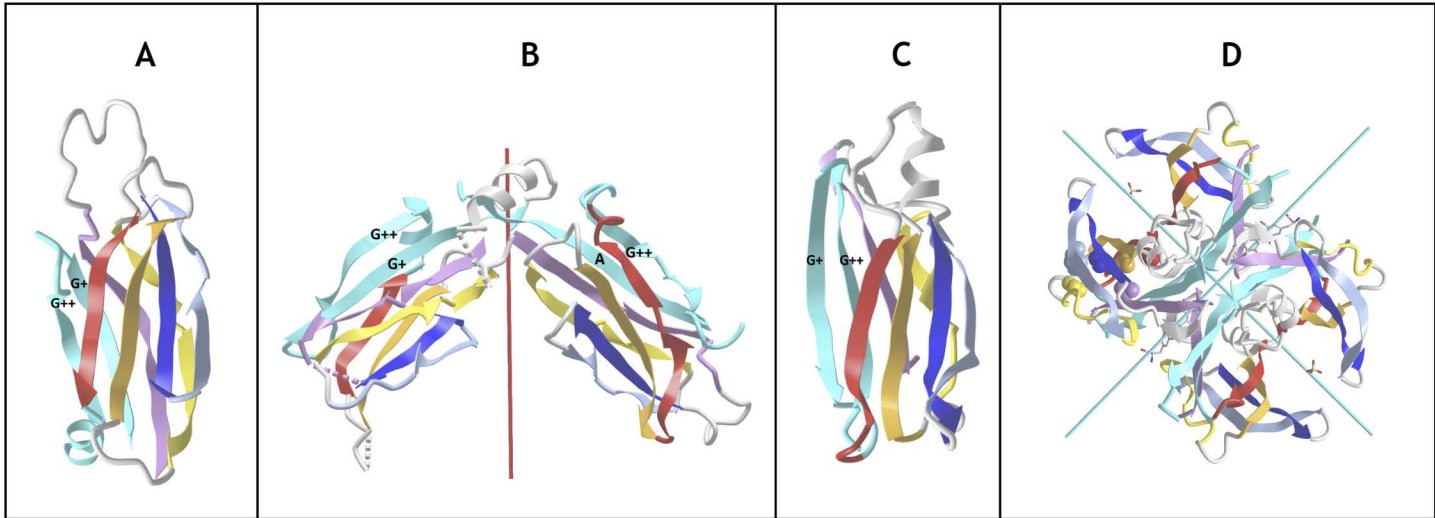

**Fig 7. Ig domain extension and circular permutations A) C2A domain - This type II C2 domain has a G+G++ hairpin extension (shown in cyan) and no A.** The G++ is a circular permutation of the A strand of a Type I C2 domain. B) **Synaptotagmin 2 tandem domains C2A - C2B** (PDBid: 4P42) in a C2 pseudo-symmetric head to head tandem arrangement. The C2B domains the A strand circularly permuted w.r.t. G++ strand in C2A (A and G+ in cyan) https://structure.ncbi.nlm.nih.gov/icn3d/share.html?T2NCEnWVHaoW2AE9A&t=4P42. C) **Transthyretyin (PDBid 4P42) domain. Similarly it shows a G+G++ extension but in a reverse order vs. C2A forming a parallel G++ strand to the B strand** https://structure.ncbi.nlm.nih.gov/icn3d/share.html?wgNVJar5M1ZBwiuL9&t=2ROX. D) **Transthyretin tetramer with a D2 symmetry** formed by the G+G++ hairpin.

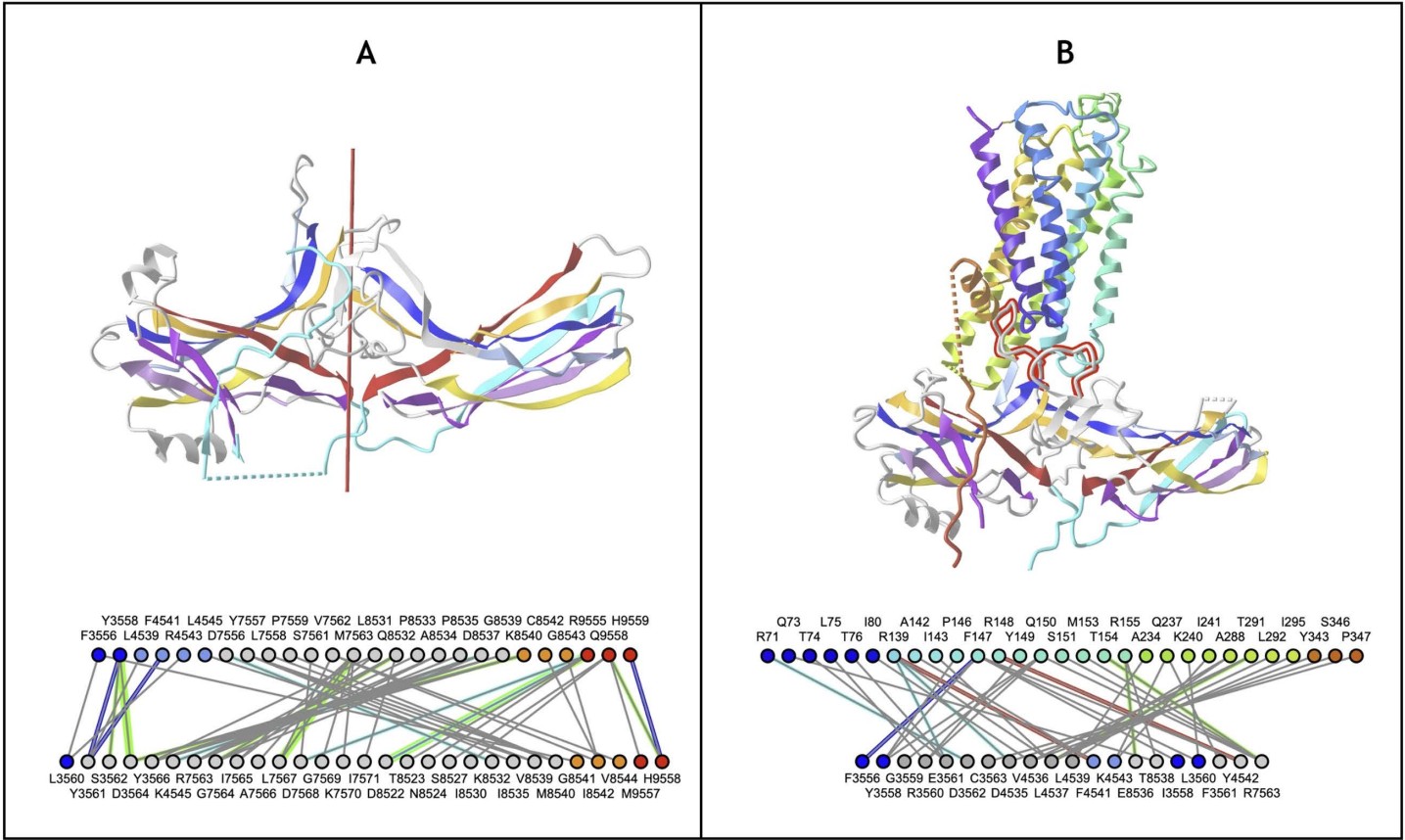

**Fig 8. Head-to-Head tandem Arrestin Structure A) Arrestin (PDBid: 1CF1)** https://structure.ncbi.nlm.nih.gov/icn3d/share.html?1LKCe6fyS1fzacVL7&t=1CF1 **B) Arrestin and GPCR 7-TM domain interacting in an active conformation** (PDBid: 6TKO). The Arrestin_N domain's Finger loop (CC') and Middle loop (extended EF loop) interact with the transmembrane helices of activated G-protein-coupled receptors. The additional A- strand (see text) is shown in cyan to highlight its role in enabling a head to head configuration between the two Arrestin Ig domains. https://structure.ncbi.nlm.nih.gov/icn3d/share.html?He7U1h5mUbipbVW58&t=6TKO.

It is important to note that the "head" we consider here is formed by the CC' and EF loops. When compared to canonical Ig domains, Arrestins have significantly extended loops that are key to their function in binding GPCRs. In the Arrestin_N domain, the Finger loop corresponds to an extended CC' loop that engages the cavity between the transmembrane helices of activated G-protein-coupled receptors [96], and the Middle loop corresponds to an extended EF loop (see Fig 8B). These two loops correspond respectively to what could be called the anti-CDR3 (CC') and the anti-CDR1 loop (EF) that are positioned pseudo-symmetrically to FG (CDR3) and BC (CDR1) loops in antibodies [61].

**Lamins.** Lamins are thought to be present only in the animal kingdom. They are Intermediate filament proteins that make up the nuclear lamina and are involved in diverse functions such as chromatin organization, gene regulation, and cell differentiation [97]. There are three Lamin genes in the human genome: Lamin B1, Lamin B2, and Prelamin A/C. The latter undergoes post-translational processing into Prelamin A or Prelamin C as the two major forms. Human Lamins contain an Ig-like domain in the tail region, the LTD (Lamin Tail Domain) exhibiting several unique structural features with respect to a typical Ig domain.

Lamin Tail domains (LTD) contain 9 strands **A⁻ABCC'EE⁺FG** that differ in part from canonical IgV or IgC domains (Fig 9). They form a beta sandwich with two sheets **with a highly unusual Sheet1 formed by ABEE⁺**, lacking the antiparallel D strand (in the usual

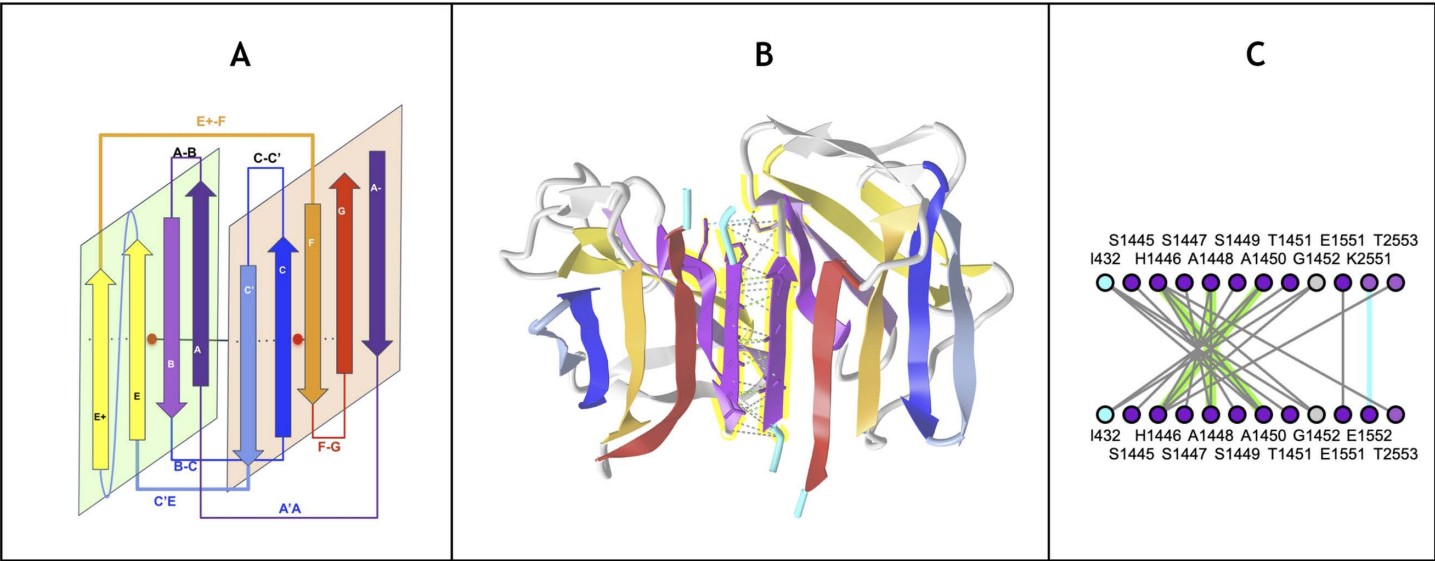

**Fig 9. Example of the Ig-extended Lamin Tail Domain (LTD).** A) Schematic LTD strand topology forming sheets ABED+ and A-GFCC'. B) **Pseudo symmetric LTD dimer** brings the two A-GFCC' as one in antiparallel through the additional A- strand at the N terminus. C) **LTD dimerization interface** using the IgStrand numbers for the A- strand (i=1; j=4 running from 1445 to 1452) and a few residues on strand A and B. Note that the current IgStrand numbering does not cover residues before 1445 (based on current template - see Table 1) https://structure.ncbi.nlm.nih.gov/icn3d/share.html?BRks581vEqzUpENs9&t=7DTG.

ABED Sheet1 encountered in IgV and especially IgC domains) but containing an E+ strand instead, following the E strand and running parallel. **Sheet2** on the other hand is formed as **A-GFCC' strands where the G strand is flanked by the additional A- strand in antiparallel instead of the parallel A' strand found in IgI/IgV domains (A'GFCC')**. Topologically, apart from the E+ strand insertion, it is similar to FN3 domains that also possess no D strand and form a **C'E cross-β sheet loop**. However, the peculiar insertion of an E+ strand gives rise to a **unique E+F cross-β sheet loop**, instead of the canonical EF loop, that forms a beta strand and antiparallel 2-stranded beta sheet with the BC loop (CDR1) also forming a beta strand, a very unique structural region of Lamins, implicated in Lamins specific functions and where mutations cause a number of diseases [97,98]. The Ig hallmark of the Ig-fold consists in the four central beta-strands B,C,E,F, forming two loops straddling the two sheets of the barrel BC (CDR1) and EF. Lamin's E+ insert displaces the latter as a **E+F loop**, a notable difference w.r.t. other Igs that may give it specific structure-function specificity; but conserving the GFCC' sheet with CC' and FG(CDR3) loops as in classical IgV antibodies.

The presence of the additional A- strands on the N-terminus sandwich barrel edge offers a side-by-side dimerization interface, observed in experimental structures of individual Lamin Tail domains (LTD) (PDB: 3UMN;7DTG in Lamin B1) [99,100] with the formation of backbone hydrogen bonding of the A- strands of each monomer in antiparallel.

**Transcription factors Ig-domains.** Some transcription factor domains exhibit Ig-like and Ig-extended domains. This is the case for tumor suppressor p53 (Fig 10A) that possess an N-terminal A+ strand extension between A and A' and an insertion that forms two extra strands, one contributing to the ABE strand in position of a D strand and one contributing to the GFCC' sheet in a C" position before the C' strand (see Fig 10). Similarly, the Rel Homology Domain (RHD) is found in eukaryotic transcription factors, in particular in NF-κB and NFAT. It exhibits an extension forming two strands as in p53 between C and C'. Its N-terminus however is similar to IgV domains forming a split AA' strand. It is followed by another Ig-C2 like domain, the IPT (Immunoglobulin, Plexin, Transcription factor) domain,

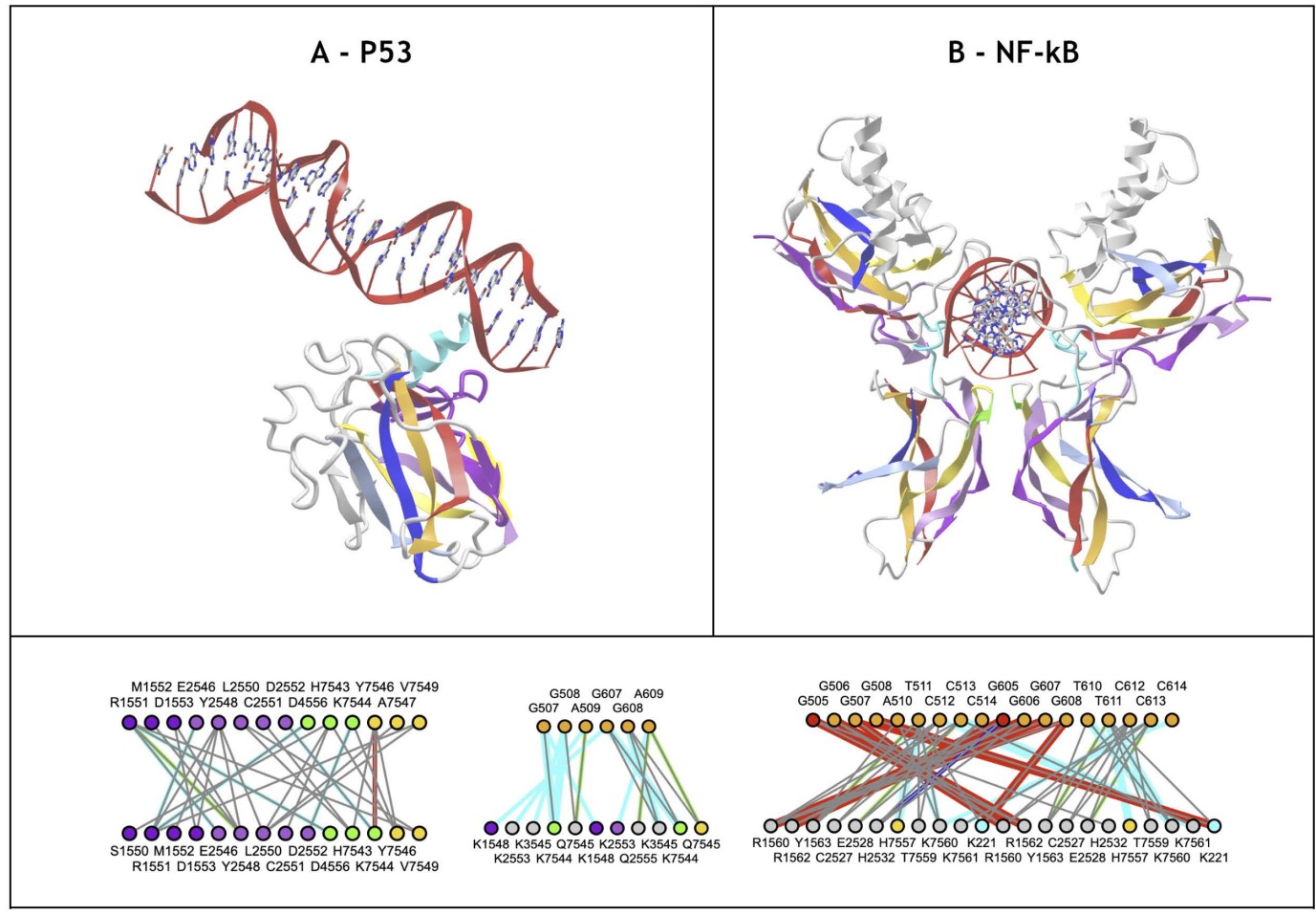

**Fig 10. Transcription factors Ig-extended and Ig-like domains.** A) **NF-kB RHD and NF-kB IPT dimer.** The Rel Homology Domain (RHD) in tandem with the Immunoglobulin-Plexin-Transcription factor (IPT) domain form a dimer that binds DNA https://structure.ncbi.nlm.nih.gov/icn3d/share.html?NVxjekmS-MxWnrs5P7&t=1A3Q. The RHD exhibits an insertion between strands C and C' extending the ABE strand in position of a D strand and one contributing to the GFCC' sheet in a C" position before the C' strand in that order. The linker region between C and E strands is therefore inverted as compared to the canonical IgV domains. If we name strands according to the IgV canonical strand names, the topology runs C>DC"C'>E in RHD as opposed to C>C'C"D>E in IgV. Overall the topology can be seen as a permuted IgV-like topology, forming two sheets GFCC'C" and ABED as in IgV domains. It is followed by the IPT (Immunoglobulin, Plexin, Transcription factor) domain, also known as TIG (Transcription factor ImmunoGlobulin), with an IgC2 topology extended by an unusual small D strand and a split AA' N-Terminal strand. The two IPT domains dimerize through their ABED sheet, in parallel. The two chains bind DNA pseudo-symmetrically. B) **P53 DNA Binding Domain** possesses, as in NF-kB RHD, a permuted IgV-like topology https://structure.ncbi.nlm.nih.gov/icn3d/share.html?RiTYSGVRTWCA2AyZ8&t=1TUP with an additional N-terminal strand extension before and between A and A'. **C) Residue Interactions** in the IPT-IPT dimer interface, **the two IPT domains** - DNA interface, and the **two RHD domains** - DNA interface.

also known as TIG (Transcription factor ImmunoGlobulin). In the p53 DNA binding domain or the NFAT or NF-kB Rel homology domain the linker region between C and E strands is inverted as compared to the canonical IgV domains: if we name strands according to their positions in the latter, the topology runs C>DC"C'>E as opposed to C>C'C"D>E, leading to a permuted IgV-like domain.

**Ig-domains in viruses.** Ig-like domains are found in numerous viruses. This has been largely described in the literature [3,101]. Fig 11A shows the example of the murine leukemia virus' RBD (Receptor Binding Domain) (PDBid: 1AOL), responsible for binding the murine PIT2 receptor, that exhibits an extended Ig-domain framework with receptor binding regions called VRA and VRB correspond to loops BC (CDR1) and DE (HV4) in canonical

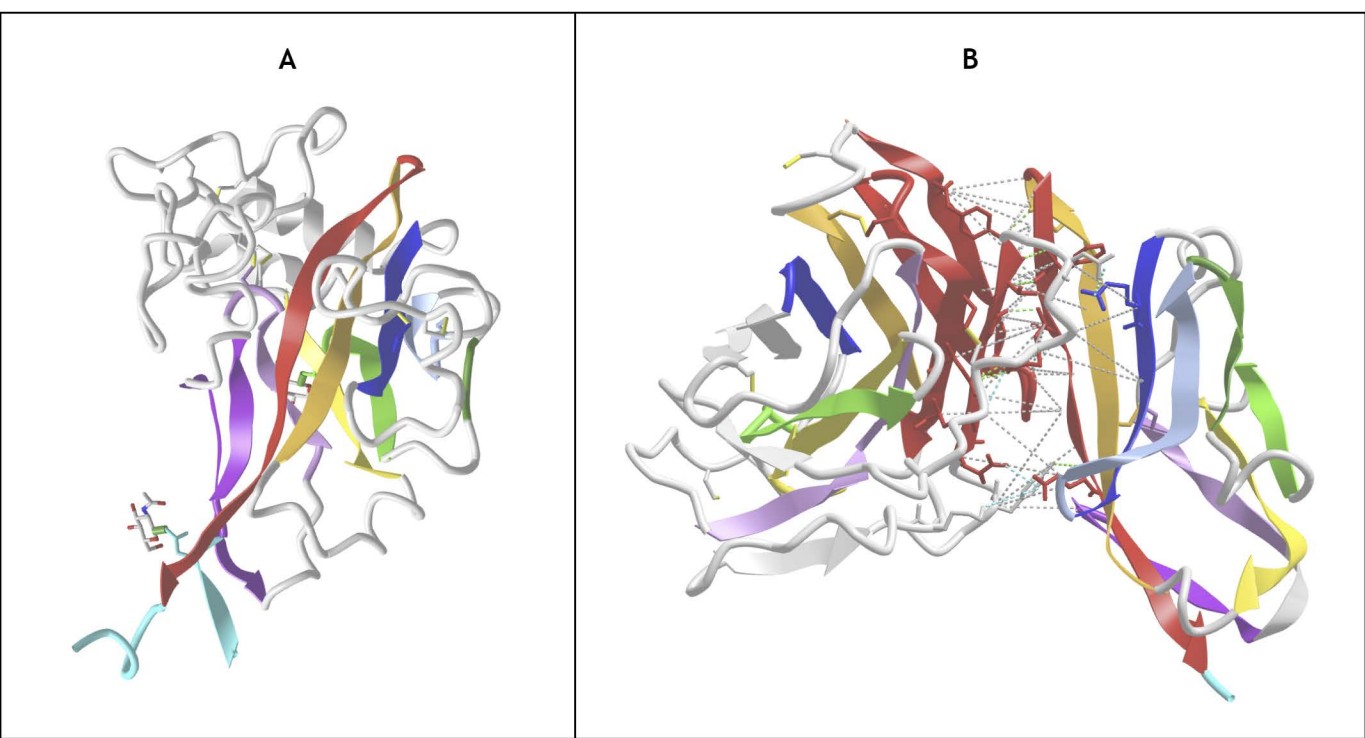

**Fig 11. A) The Murine Leukaemia viruses envelope RBD (PDB id 1AOL) SCOP b.20 shows an extended Ig-domain framework with receptor binding regions called VRA and VRB correspond to loops BC (CDR1) and DE (HV4) in canonical Ig-domains.** The Ig-domain is extended at the N-terminus with a A'- strand hydrogen bonded to the G-strand in antiparallel and forming a A'-A' hairpin extending the A'GFCC'C" to A'A-GFCC'C" where the A'- in inserted between the A' and G strand https://structure.ncbi.nlm.nih.gov/icn3d/share.html?hA7Ftj6A2QEMoCSV7&t=1AOL. The leukemia virus envelopes can be compared to the Feline Leukemia virus RBD as well as the endo-retrovirus EnvP(b)1 https://structure.ncbi.nlm.nih.gov/icn3d/share.html?oFgS8ac-CM7AW1hZDA&t=6W5Y,1LCS,1AOL. B) **The monkeypox virus protein M2 binding CD80.** (PDBid: 8HXA). This beta sandwich is similar to the SECRET domain found in other orthopoxviruses binding chemokines. This domain is also called PIE (Poxvirus immune evasion). In this case, the M2 protein modulates T cell co-stimulation in binding CD80. It is missing the A strand altogether at the N terminus. It also presents an insertion of two strands between B and C forming a hairpin C--C-, that take the place of C" and C' (in reverse order vs. a canonical IgV domain), extending Sheet 2. The C-terminal extension is composed of 3 additional strands to the G-strand, first a G+ intercalating strand forming a GG+ hairpin and a parallel beta sheet interface to strand F, extending the GG+FCC' followed by a G++G+++ hairpin extending Sheet1 as G+++G++BED, the latter G+++ strand replacing the missing A strand, similarly to a circular permutation as found in type **II** C2 domains. The M2 domain binds the G-strand (red) of CD80 laterally through its C-terminus G-strands extension (red), mainly G++/ G+++, to modulate T cell co-stimulation. https://structure.ncbi.nlm.nih.gov/icn3d/share.html?cbid7J2RCJhMNRca8&t=8HXA [110].

Ig-domains. The Ig-domain is extended at the N-terminus with an additional A+ strand hydrogen bonded to the G-strand in antiparallel (insertion between A' and G) extending the GFCC'C" sheet [102,103] to A+A'GFCC'C" of an otherwise IgV topology. While Ig-related genes have been acquired by viruses from the ancient genomes, it is interesting to note that integration in the human genome of an estimated 8% of sequences of retroviral origin (HERV) that are remnants of ancient retroviral endogenization. Among them, the endogenous retroviral envelope EnvP(b)1 protein is coded by a gene found in human and other primate genomes, and traced to an event estimated between 40 and 71 million years ago [104,105]. The receptor binding domains (RBD) of human EnvP(b)1 structure (PDBid: 6W5Y) exhibits an IgV topology (A'GFCC'C" - ABED) that defines structural similarities with extant leukemia viruses' envelopes, the main difference lying in the presence of the additional A- strand in the latter, pointing to a deletion or addition of that strand in EnvP(b)1 during its evolution.

Viral pathogens have evolved sophisticated mechanisms to evade the host immune system, this is the case for example of orthopoxviruses [106,107] that contain poxvirus immune

evasion proteins (PIE) [108]. PIE domains can be considered Ig-extended. For example, the SECRET domain (smallpox virus-encoded chemokine receptor) represents a family of viral CKBPs (chemokine-binding proteins) that modulate the chemokine network in host response in binding chemokines to inhibit their activities [109]. The recently determined monkeypox virus M2 protein structure (Fig 11B) shows how it interacts with the immune system [110], with an exquisite structural interface between the Ig-extended MPXV M2 protein and the extracellular variable Ig-domain of CD80 (Fig 11B) that interrupts the natural interactions of human B7.1/2 with CD28 and CTLA4 and subverts T cell activation mediated by B7.1/2 costimulatory signals.

### Ig or not Ig? Blurring (or swapping) the limits of the Ig-fold

While our IgStrand numbering scheme is aimed at representing all domain variants of an Ig-fold, it can apprehend similarities with other beta sandwiches, using the same strand nomenclature. Any Ig-domain variant contains the full 5-strand core supersecondary structure formed by B-C, E-F, plus either an A, A' or split A/A' strand at the N-terminus, a central linker between C-E strands with one or more strands among C',C'',D, and the G-strand at the C-terminus and we defined Ig-extended domains as encompassing the Ig-fold irreducible signature with additional strands as explained earlier (Fig 2).

**Strand swaps** can also be considered and can relate diverse variants of Ig-like vs. beta sandwiches classified as different folds, such as jelly rolls [75]. Beta sandwiches are estimated to represent overall 30% of all folds composed of beta strands [76]. However, among these, after the Ig-fold, Jelly-rolls represent the second major beta sandwich fold. In terms of numbers, taxonomy databases classify dozens of beta sandwich folds, for example CATH currently counts 44 of them representing 536 superfamilies for a total number of 60,890 domains known experimentally. The Ig fold alone accounts for 330 superfamilies (62%) and 45,422 domains, while the Jelly-roll accounts for 121 superfamilies (22%) and 11,770 domains, the two combined representing 84% of beta sandwich superfamilies, dwarfing the other 42 folds combined that represent only 85 superfamilies (16%) and 3,698 domains known experimentally. These numbers can now be re-examined with predicted structural proteomes becoming available, and work is underway on the Taxonomy of the Alphafold database [111,112]. Focusing on the Ig-fold and Jelly Roll fold, it is clear that evolution has favored these two main folds to perform a myriad of biological functions. One can relate the two folds topologies by considering a conceptual concerted strand swap of the A and G strands at the N- and C-terminus, respectively (Fig 12).

### Three levels of organized complexity: Ig domains, chains, assemblies

We can consider three levels of complexity: at the tertiary domain, tertiary chain, and quaternary assembly. Many protein chains contain Ig- or Ig-like domains. When considering the Ig-fold plasticity in light of extreme diversity of molecular surfaces of interaction they can form, it is no surprise, but awe, that the hierarchical complexity of tertiary Ig-chains and their quaternary assemblies can orchestrate complex systems in vertebrate organisms. 3D structures start to emerge for proteins such as Pom210 with Ig-chains counting 17 Ig-domains in tandem in the context of the full nuclear pore receptor complex and allow to visualize conformational plasticity of Ig-chains [113,114].

In the cell surface proteome, or surfaceome, a high number of cell surface receptors responsible for cell-cell interactions at the heart of the immune system, the nervous system, the vascular system (see examples in Fig 1), the extracellular regions (ectodomain) of single pass cell surface proteins form Ig-chains that can contain from one to tens of Ig domains,

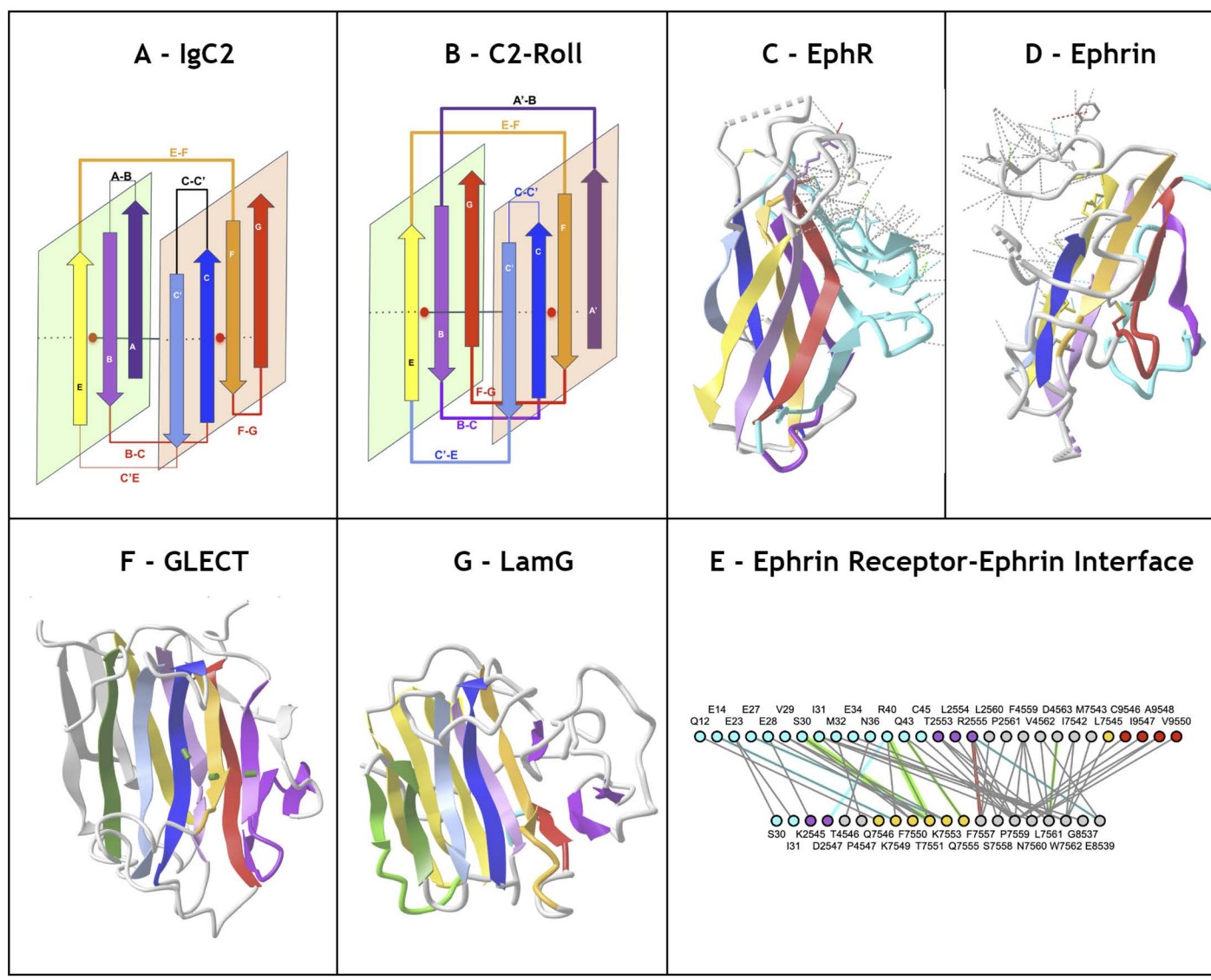

**Fig 12. Ig-like and Jelly-roll folds. A) IgC2 schematic topology B) Conceptual A and G strands swap comparing Jelly-roll to Ig-like** transforming AB and GF loops into straddling loops between Sheets 1 and 2 of the sandwich changing from ABE and GFCC' in an IgC2 domain to BGE and AFCC' sheet respectively in what could therefore be called a C2-roll domain, using the Ig strands nomenclature. **C) The Ephrin Receptor** (PDBid: 3GXU), a **Galactose-binding domain-like fold** classified as a jelly-roll in CATH, and ECOD and SCOP (**b.18**) represents a good example of an extended "C2-roll" domain where the N-terminus provides an additional set of strands, extending both sheets and providing the bulk of the interface its ligand Ephrin. **D) Ephrin** exhibits an Ig-extended **Cupredoxin-like** beta sandwich fold (SCOP **b.6**). The G-strand is forming, as in many IgV domains, an FG loop and a parallel A' strand (A'GFCC') but it is preceded by an N-terminus extension (in cyan) providing additional strands A- on Sheet 2 to form A-A'GFCC' and a strand A-- on Sheet 1 to form the A--BE, the interacting surface to the Ephrin Receptor https://structure.ncbi.nlm.nih.gov/icn3d/share.html?kRY8dJJvV4aM7fFx6&t=3GXU. **E) The Ephrin Receptor-Ephrin interactome. F) Galectin (GLECT) domain** is a galactose-binding lectin (PDBid 4AGV) that represents **yet another Jelly-roll variant topology,** classified as a **Concanavalin A-like** (SCOP **b.29**). GLECT exhibits the A and G strand swap between the sandwich sheets as compared to an IgV domain, extending one sheet with an additional C''' strand (in grey) after C'' to form a sheet AFCC'C''C''', and with a two strand E+E++ β-hairpin insertion (in grey) after E to form the GBEE+E++ extended sheet. https://structure.ncbi.nlm.nih.gov/icn3d/share.html?rYBVVeHiBh2TkPVM6&t=4AGV. **G) Laminin G-like (LamG) domain i**s also classified as a **Concanavalin A-like jelly roll** with a two strands extension at the N-terminus extending the first sheet A--A-GBEE+E++ as compared as to GLECT https://structure.ncbi.nlm.nih.gov/icn3d/share.html?bmYk9ciuwBQb2td47&t=5DZE.

and even in hundreds in the muscular system. CD8 for example contains one extracellular Ig domain, CD4 contains four of them, CD22 has seven, cell adhesion proteins such as DSCAM (see later) can contain up to 16 Ig-like domains, including FN3 domains. Much longer chains, such as the giant protein Titin can contain hundreds of chained Ig and FN3 domains [115]. Considering alternative splicing of Ig-domain containing genes can produce a myriad of proteins with variable numbers of Ig domains, adding more complexity due to the already large number of Ig domain variants, a bewildering number of Ig-domain combinations in Ig-chains (see for example [116,117]), making the Ig-proteome very large and diverse.

**Self assembly of Ig-domains.** Considering the omnipresence of Ig-domains among cell surface receptors, especially in the immune system and the nervous system, one is naturally led to consider their role in cell communication. This is a vast subject, but we only want to focus on the contacting mechanisms of these molecules at the surface of cells that may lead to attractive or repulsive actions involved in synapse formation. In neuronal synaptogenesis, the primary cells involved are neurons, which form synaptic connections with other neurons or target cells [17–19,118]. In immunological synaptogenesis, various immune cells such as T cells, B cells, dendritic cells, and macrophages, form contacts with each other or with non-immune cells such as APCs or target cells.

A striking feature of Ig domains at the surface of cells is their pairing, in both cis and trans. External residues on the surface of Ig domains themselves represent a 3D scaffold for Ig-recognition. Yet that recognition, both attractive and repulsive at the heart of self- vs. non-self, is based on co-evolved codes between external residues between Ig-receptors and Ig-ligands (see for example [119]). The combinatorics of 3D Ig-pairing for a given Ig-domain variant type such as IgV domains, found in so many cell surface receptor-ligands interfaces, is staggering. Multi-Ig-domain chains bring even greater combinatorial capabilities [14,120–123]. When one considers the diversity of Ig-domain topo-structural variant types and/or Ig-chain constructs with alternative splicing, the conceptual pairwise interactome numbers become impossible to grasp.

While a myriad of distinct Ig-Ig interactomes discriminate self- vs. non-self between cells, a number of Ig-Ig interactomes have been determined experimentally leading to the observation of conserved Ig-Ig pairwise interaction interface structures that can in some cases be seen as "quaternary folds". This is the case of IgV-IgV canonical interfaces found in antibodies, TCRs but also in many cell surface receptor ligands of the immune system or the nervous system, some being common to both such as nectins. An initial survey of the Protein Data Bank (PDB) detected the presence of at least one Ig-domain in 16,661 protein structures that included 28,795 pairs of Ig-domains in contact, with 24,222 with high confidence, as identified through the IgStrand algorithm (see methods).

The IgStrand based Ig-detection algorithm is highly reliable in detecting domains with an Ig-fold; but it does not accurately label the variant types (this is a function of the structural fit with templates (Table 1). Work is underway to improve this accuracy and determine the diverse classes of interfaces (or quaternary folds) yet an early analysis using CDDs tend to show that Ig domains pair according to their types, i.e., IgV-IgV, IgI-IgI, IgC1-IgC1, IgC2-IgC2.

**Canonical quaternary antibody IgVH-IgVL and IgCH1-IgCL interactomes.** Antibody heavy and light chains assemble through both i) VH and VL domains using the [A']GFCC'[C"] sheet in parallel, and ii) CH1 and CL domains using the ABED sheet in antiparallel, respectively (see schematic Fig 13). The paired variable and constant domains form interfaces VH:VL and CH1:CL interfaces that can be considered canonical. The affinity and specificity of an antibody towards an antigen results from diverse factors among which the pairing and dynamics of VH:VL and CH1:CL domain quaternary interfaces and their relative

## A - IgVH-IgVL parallel

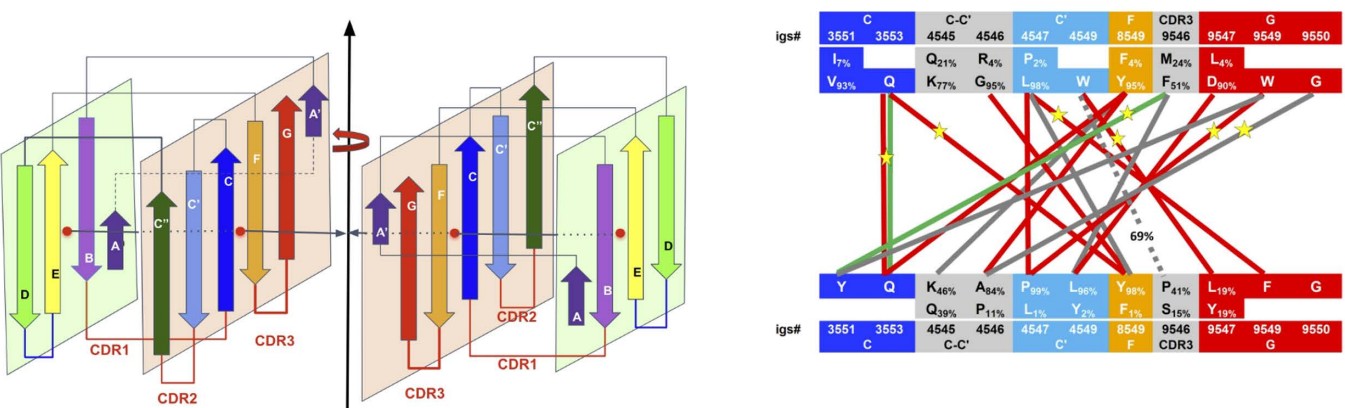

## B - IgCH1-IgCL antiparallel

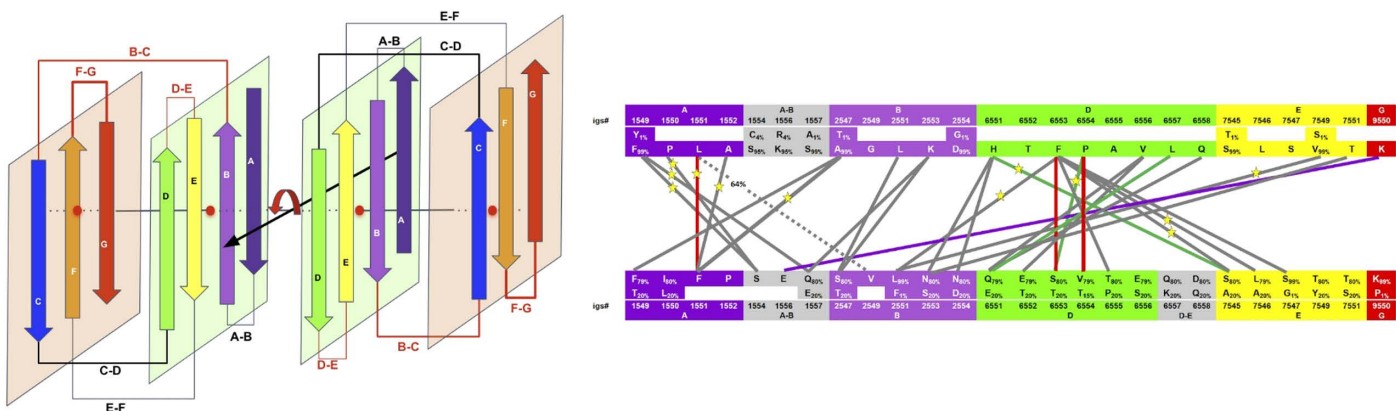

**Fig 13. Schematic representation of Ig-domains dimerization patterns and contact network A) VH:VL schematic parallel interface and conserved interactions B) CH1:CL schematic antiparallel interface and conserved interactions (see text).** On the left a schematic representation of the domain interface with the C2 pseudo symmetry axis. Note that the symmetry axis for VH:VL is vertical corresponding to a parallel interface, while it is horizontal for CH1:CL as the interface is antiparallel. On the right the residue interaction network. Residue subscripts indicate the percentage of occurrence of that residue at that position, if less than 100% in the dataset. Ig strands are color-coded according to the iCn3D IgStrand scheme (See Methods). Only residue-residue interactions present in 70% of the Fabs in each dataset are shown with solid lines. Dotted lines with a % number represent interactions that are present in at least 70% of the Fabs in the diverse antigen binding dataset, with the % number representing the % of Fabs in which this contact is present in the SARS-CoV-2 antigen binding dataset. Interaction lines between residues in VH:VL and CH1:CL domains are color-coded in red for symmetric contacts, green line for hydrogen bonds and purple for a noticeable conserved ionic interaction in CH1:CL; otherwise, grey indicates van der Waals interactions. The highly common interactions, present in at least 90% of the dataset, are indicated by a star. See Tables 2 and 3 for more details. The VH:VL and CH1:CL interfaces of a specific Fab (PDBid:7LM8) can be visualized and analyzed with the iCn3D link: https://www.ncbi.nlm.nih.gov/Structure/icn3d/share.html?uvawHK16NinhaZL68.

variable-constant two-domain interfaces and elbow angle [124–126]. The dynamics of the interface can be measured by the elbow angle, which is the angle formed by the pseudo-2-fold axis connecting CH1 to CL and VH to VL. This angle is dependent on various combinations of germline heavy and light chains [127,128]. The residues on the VH-VL interface can influence the VH-VL packing geometry and the characterization of VH-VL and CH1-CL interfaces is important for understanding the function of antibodies. Here, the IgStrand numbering scheme is used to analyze interactions between VH:VL and CH1:CL Ig-domains.

**Ig-Ig domain interfaces can be compared across a dataset of any size and across any Ig-fold variants using the Ig-Strand numbers**. This is naturally true for variable and constant domains of antibodies. Given the importance of antibodies, we cover in this section some elements of the interfaces between the variable domains and the constant domains of the heavy and light chains that constitute the majority of known antibody fragment structures. To do so, we selected two datasets of the same size: 1) a first dataset comprising **107 Fab antibody fragments** binding to the SARS-CoV-2 spike protein, referred to as **the SARS-CoV-2 antigen binding dataset**. They were contained in 76 PDB structures and selected with a resolution lower than 2.75Å from a larger ensemble of 258 Fab antibody fragments contained in 165 PDB structures; 2) a second dataset comprising 107 Fab antibody fragments binding to diverse antigens, referred to as **the diverse antigen binding dataset**. They were contained in 107 PDB structures, randomly chosen with a resolution lower than 2Å. The aim being to survey VH:VL and CH1:CL interfaces in terms of IgStrand residue numbers and their possible dependence on a diverse vs. specific antigen binding. Table A in S1 Text summarizes the number of contacts seen in these datasets. The details of these datasets, the IgStrand based sequence alignments for the VH, VL, CH1, CL domains, and the VH:VL and CH1:CL interactomes for both datasets are available in Supplement files S8 Data and S9 Data.

Fig 13, Tables 2 and 3 present the common VH:VL and CH1:CL interactomes in the SARS-CoV-2 antigen binding dataset. Tables B and C in S1 Text present the same for the diverse antigen binding dataset. Note that contact analyses using IgStrand numbers are alignment independent: the individual VH, VL, CH1, and CL domains are mapped to corresponding Fab templates (Table 1) to obtain reference IgStrand numbers and it follows that contacts inherit IgStrand numbers. No multiple alignment in sequence or structure is needed. They are aligned through the IgStrand numbering: once antibody domain structures are aligned to templates and assigned IgStrand numbers, they are all aligned by transitivity. In this first application of the IgStrand numbering scheme to antibodies, static structures of Fabs determined experimentally and obtained from the PDB database are used. Residue-residue contacts

**Table 2. VH:VL interactions of Fabs binding SARS-CoV-2 spike protein** (70% cutoff). Red numbers represent symmetric contacts. Bold contacts represent highly conserved hydrogen bonding contacts. Shaded cells represent five highly conserved contacts (90% cutoff) shared between SARS-CoV-2 antigen binding dataset and diverse antigen binding dataset (Table B in S1 Text).

| VH igs# | VL igs# | VH<>VL | #Fabs | %Fabs | contacts | hbonds | ionic | pi-stacking | pi-cation |
|---------|---------|--------|-------|-------|----------|--------|-------|-------------|-----------|
| 4547 | 9549 | L/P<>F | 107 | 100 | 107 | | | | |
| **3553** | **3553** | **Q<>Q** | 107 | 100 | 107 | **105** | | | |
| 4549 | 9547 | W<>L/Y | 103 | 96 | 103 | 3 | | 36 | 7 |
| 9549 | 4547 | W<>P/L | 102 | 95 | 102 | | | | |
| 3553 | 8549 | Q<>Y/F | 97 | 91 | 97 | | | | |
| 9550 | 4546 | G<>A/P | 97 | 91 | 97 | 3 | | | |
| **9546** | **3551** | **F/M<>Y** | 96 | 90 | 96 | **80** | | 1 | |
| 9549 | 3551 | W<>Y | 94 | 88 | 94 | 2 | | 24 | |
| 4547 | 8549 | L/P<>Y/F | 94 | 88 | 94 | | | | |
| 8549 | 4546 | Y/F<>A/P | 92 | 86 | 92 | | | | |
| 9546 | 4549 | F/M<>L/Y | 92 | 86 | 92 | | | | |
| 8549 | 3553 | Y/F<>Q | 90 | 84 | 90 | | | | |
| 4547 | 4547 | L/P<>P/L | 89 | 83 | 89 | | | | |
| 4546 | 8549 | G/R<>Y/F | 87 | 81 | 87 | | | | |
| 9547 | 4549 | D/L<>L/Y | 82 | 77 | 82 | 1 | | | |
| 8549 | 4545 | Y/F<>K/Q | 79 | 74 | 79 | | | | |

**Table 3. CH1:CL interactions of Fabs binding SARS-CoV-2 spike protein (70% cutoff).** Red numbers represent symmetric contacts. Bold contacts show highly conserved hydrogen bonding. Bold contacts show highly conserved hydrogen bonding, and in purple for a highly conserved ionic contact. Shaded cells represent twelve highly conserved contacts (90% cutoff) shared between SARS-CoV-2 antigen binding dataset and diverse antigen binding dataset (Table C in S1 Text).

| CH1 igs# | CL igs# | CH1≤≥CL | #Fabs | % Fabs | contacts | hbonds | ionic | pi-stacking | pi-cation |
|---|---|---|---|---|---|---|---|---|---|
| 6553 | 7546 | F<>L/A | 107 | 100 | 107 | | | | |
| 1552 | 1551 | A<>F | 104 | 97 | 104 | | | | |
| 6553 | 2551 | F<>L/F | 104 | 97 | 104 | | | 1 | |
| **6551** | **7545** | **H<>S/A** | 103 | 96 | 103 | **81** | | | |
| 2547 | 1551 | A/T<>F | 103 | 96 | 103 | | | | |
| 1549 | 1557 | F/Y<>Q/E | 103 | 96 | 103 | | | | |
| 1550 | 1554 | P<>S | 102 | 95 | 102 | 2 | | | |
| 1551 | 1551 | L<>F | 102 | 95 | 102 | | | | |
| 7549 | 2551 | V/S<>L/F | 101 | 94 | 101 | | | | |
| **6554** | **6553** | **P<>S/T** | 100 | 93 | 100 | **63** | | | |
| 6553 | 7545 | F<>S/A | 100 | 93 | 100 | | | | |
| 1549 | 1554 | F/Y<>S | 98 | 92 | 98 | | | | |
| 6553 | 7547 | F<>S/G | 94 | 88 | 94 | | | | |
| 6556 | 6551 | V<>Q/E | 94 | 88 | 94 | | | | |
| 2553 | 2547 | K<>S/T | 93 | 87 | 93 | | | | |
| **9550** | **1556** | **K<>E** | 91 | 85 | 73 | 10 | **91** | | |
| 6551 | 2553 | H<>N/S | 91 | 85 | 91 | 5 | | | |
| 2551 | 2547 | L<>S/T | 89 | 83 | 89 | | | | |
| 6551 | 2554 | H<>N/D | 88 | 82 | 85 | 21 | 3 | | |
| 6558 | 6551 | Q<>Q/E | 87 | 81 | 87 | | | | |
| 6554 | 6554 | P<>V/T | 86 | 80 | 86 | | | | |
| 6553 | 6553 | F<>S/T | 86 | 80 | 86 | | | | |
| 6553 | 6555 | F<>T/P | 86 | 80 | 86 | | | | |
| **6557** | **6551** | **L<>Q/E** | 85 | 79 | 85 | **51** | | | |
| 2547 | 1549 | A/T<>F/T | 85 | 79 | 85 | | | | |
| 2553 | 1557 | K<>Q/E | 83 | 78 | 83 | 1 | 17 | | |
| 7551 | 2553 | T<>N/S | 81 | 76 | 81 | 1 | | | |
| 6556 | 6552 | V<>E/T | 78 | 73 | 78 | | | | |

at the interface between variable VH:VL or constant CH1:CL domains are determined using non-hydrogen atoms in those structures (since many PDB structures don't resolve hydrogen atoms) based on a distance and atom type criterion. An in-depth analysis on a large dataset is beyond the scope of this paper but will be performed in the future, the aim here being to demonstrate the use of the IgStrand numbering in the context of antibodies.

**VH:VL interface.** The canonical "quaternary fold" for VH:VL paired domains exhibits a C2-symmetric parallel 8-strand beta barrel (GFCC')*2 (see schematic in Fig 13A), where the A' and C" strands are not forming close interactions in the dimer in most cases. If we consider residue-residue contacts present in at least 70% of the structures in each dataset, the interaction analysis of the VH:VL interface identifies 16 common interacting residue pairs in the SARS-CoV-2 antigen binding Fab dataset (Tables 2 and A in S1 Text, bold lines in Fig 13A) and 14 in the diverse antigen binding Fab dataset (Table B in S1 Text). Among these, 13 contacts are shared across both datasets at the 70% threshold, yet some others are shared below that threshold, shown with dashed lines in Fig 13A (See Supplementary data SF8 and SF9 for all details). The presence of shared interacting pairs in both datasets suggests that the VH:VL interface exhibits significant conserved structural integrity independent of the antigen,

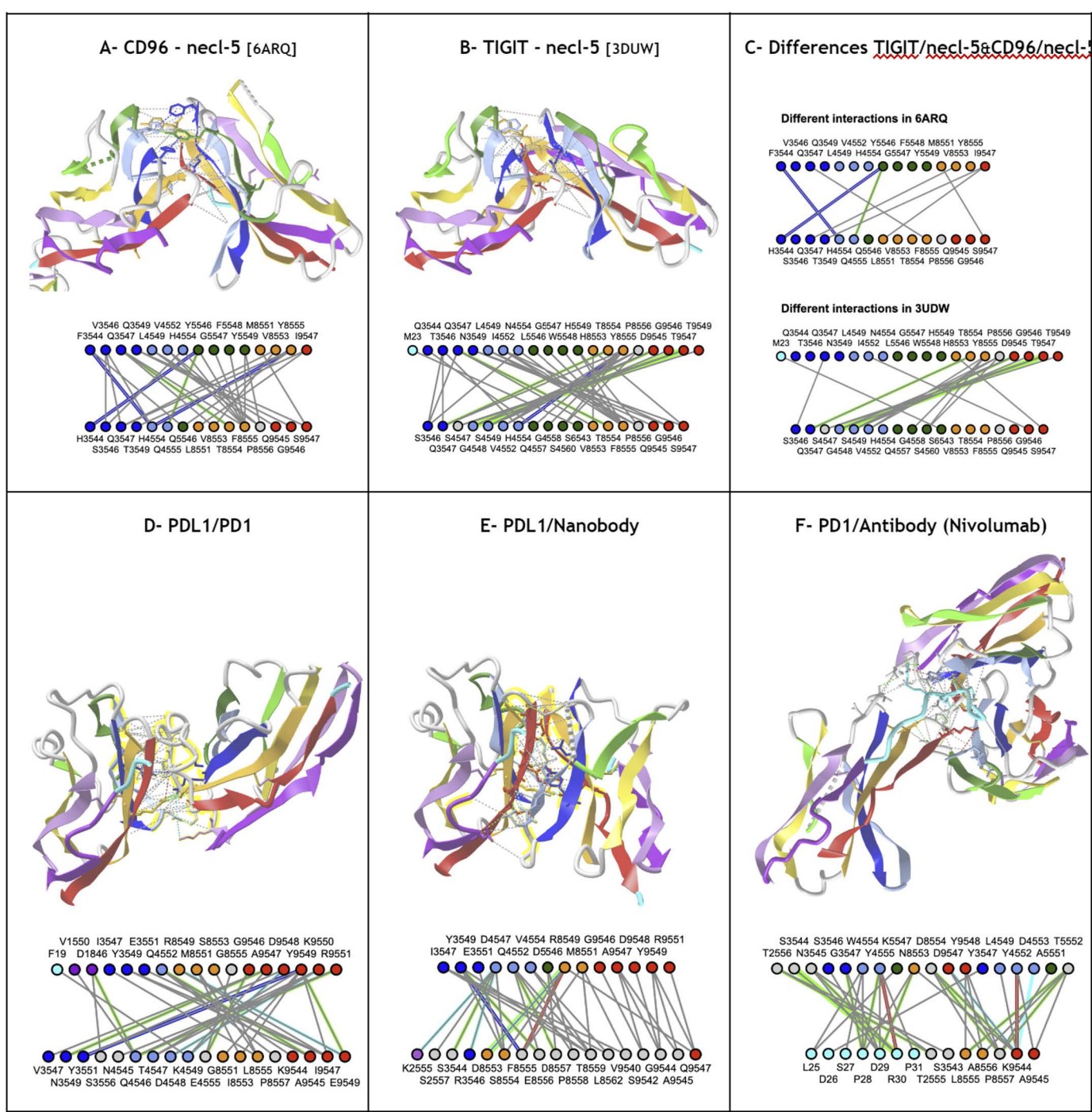

**Fig 14. Comparing IgV-IgV interfaces A-B) CD96 vs. TIGIT interacting with nectin-like protein-5 (necl-5)** (PDBid: 6ARQ, 3UDW) using their N-terminal IgV domain shown side by side with their residue interaction network, using IgStrand numbering. **C) Different residue interaction network in CD96 and in TIGIT** https://structure.ncbi.nlm.nih.gov/icn3d/share.html?HppETGxGj1yjQ7md7&t=6ARQ,3UDW. **D-E) PD-L1 interacting with PD1 (PDBid 4ZQK) vs. PDL1 interacting with a nanobody (PDBid 5JDS)** (https://structure.ncbi.nlm.nih.gov/icn3d/share.html?ehLCiHmy953yyGMeA&t=4ZQK,5JDS targeting the same epitope/surface on the PDL1 GFCC' sheet, using an elaborate FG (CDR3) loop from igs# 8553 to igs# 9547. **F) PD1/Nivolumab (PDBid: 5WT9) interaction.** Nivolumab binds to a PD1 epitope composed of the FG loop residues as well as an N-terminal loop that is considered outside of the PD1 domain in IgStrand numbering, hence the residues (in cyan)are not numbered and retain their PDB numbers. This points to the need to consider Ig domain extensions. The interface that can be compared to the PD1/PDL1 interface in D) https://structure.ncbi.nlm.nih.gov/icn3d/share.html?UhSpS48KgM7dsMUD8&t=4ZQK,5WT9.

allowing some level of plasticity. The VH:VL contacts common to at least 70% of Fabs in a dataset are listed in Table 2 for the SARS-CoV-2 antigen binding dataset and Table B in S1 Text for the diverse antigen binding dataset. Among these contacts, ten and nine contacts respectively, are symmetrically positioned (in red in Tables 2 and in B in S1 Text) with five of them (igs# 3553-3553, igs# 3553-8549/igs# 8549-3553, igs# 4547-9549/igs# 9549-4547) present in at least 90% of the structures in both datasets (shaded contacts in Tables 2 and in B in S1 Text).

Pseudo-symmetry is a remarkable property of VH and VL domain interaction, and more generally of all IgV domains presenting a canonical parallel interface [61]. As seen in Fig 13A, the side chains of the highly conserved C-strand Glutamine residue igs# 3553 (Q3553) in both VH and VL domains face each other and form a double hydrogen bond. This dimeric residue interaction is positioned on the symmetry axis and is also found in homodimers such as VL:VL and non-antibody IgV domain dimers such as CD8aa (discussed below). Mutation to Glutamate residue at that position (Q3553E) is still able to maintain one hydrogen bond, while mutation to a hydrophobic residue Valine or Leucine (Q3553V/L) abolishes the side chain hydrogen bond while maintaining a van der Waals contact, leading to more compact hydrophobic packing [129]. Another highly conserved C-strand Tyrosine residue (Y3551) in the VL domain forms hydrogen bond interactions with the Phenylalanine (F9546) at the edge of CDR3 of the VH domain in 75% of structures in the SARS-CoV-2 antigen binding Fab dataset and 62% of structures in the diverse antigen binding Fab dataset. Y3551 also forms van der Waals interactions with W9549 in the G strand of the VH domain in 88% of Fabs in both datasets. The VH residues igs# 4547, igs# 4549, and igs# 8549 along with the VL residues igs# 3551, igs# 4546, igs# 4547, igs# 4549, and igs# 8549, identified in Fig 13A, have also been found to be important in forming the VH-VL interface through covariational analysis [129]. The specific residue types may not be conserved evolutionarily but the contacts are conserved (captured by the universal IgStrand numbering) in position (the Supplementary data SF8 and SF9 lists all interactions on VH:VL domains in the two datasets). The IgStrand residue numbering provides a universal contact representation for characterizing similar interactomes across the full Ig-proteome.

**CH1:CL interface.** The CH1 domain pairs with the CL domain through a C2-pseudo symmetric interface, forming an 8-strand quaternary beta barrel (ABED)*2 in an antiparallel arrangement. This is the canonical interface between CH1 and CL domains in Fabs (see Fig 13B). The interaction analysis of the CH1:CL interface in the SARS-CoV-2 antigen binding dataset identified 28 common interacting residue pairs present (Tables 3 and in A in S1 Text, bold lines in Fig 13B) and 27 pairs in the diverse antigen binding dataset (Table C in S1 Text in at least 70% of the Fabs. Of these, 26 contacts are shared between both datasets; the ones still present in the diverse antigen binding dataset below that threshold are shown with dashed lines in Fig 13B. As in the VH:VL interface, the presence of shared interacting pairs in both datasets suggests that the CH1:CL interface provides structural integrity independent of antigen binding. The actual contacts are listed in Table 3 for the SARS-CoV-2 antigen binding dataset and Table C in S1 Text for the diverse antigen binding dataset. Among these, 12 contact pairs (igs# 1549-1554, igs# 1549-1557, igs# 1550-1554, igs# 1551-1551, igs# 1552-1551, igs# 2547-1551, igs# 6551-7545, igs# 6553-2551, igs# 6553-7545, igs# 6553-7546, igs# 6554-6553, igs# 7549-2551) are present in at least 90% of the structures in both datasets (lines with stars in Fig 13B) and share the three symmetric contacts (igs# 1551-1551, igs# 6553-6553, Igs# 6554-6554) (shown in red in Fig 13B, and Tables 3 and in C in S1 Text). The highly conserved Lysine (K9550), which is the G strand anchor (ig# 9550) in CH1, forms a conserved ionic interaction with the Glutamate residue (E1556) in the A-B loop of the CL domain in both datasets (shown in purple in Fig 13B and Tables 3 and C in S1 Text ). Side chains of highly

conserved CH1 residues igs# 6551 and igs# 6554 form hydrogen bonds with CL residues igs# 7545 and igs# 6553 respectively in 95% of the structures in the SARS-CoV-2 antigen binding dataset and in 80% of the structures in the diverse antigen binding dataset. The Phenylalanine (F1551) in the A strand of the CL domain forms three highly conserved hydrophobic contacts with three highly conserved residues in the CH1 domain: L1551, its pseudo-symmetric cognate, A1552 in the A-strand, and A2547 in the B-strand. These hydrophobic interactions are conserved in at least 90% of the structures and have been shown to provide stability to the CH1-CL dimer [130].

A continuous A strand (no A') and a D strand on the ABED sheet are two hallmarks of C1-set domains, and show a very high structure and sequence conservation (see Fig 13B). These strands are key to the dimerization of CH1:CL domains with the C2-symmetry axis positioned between contacts igs# 1551-1551 and igs# 6553-6553. The interface counts 8 D-strand:D-strand interactions, while the CH1 D-strand adds another 7 contacts with the B and E strands of the CL domain, hence a total of 15, i.e., more than half of all interactions between CH1 and CL domains, making the CH1 D-strand critical to the assembly of CH1 and CL domains. The CH1 D-strand is highly conserved in sequence (igs# 6551-6558 HTF-PAVLQ) with the highly conserved Phenylalanine residue (F6553) alone forming six contacts with CL domain residues (see Fig 13B and Table 3).

In summary, the two datasets show a consistent residue interaction network between VH and VL domains with 13 common contact pairs for VH:VL above the 70% threshold (see Tables 2, 3, A, B, and C in S1 Text). Interestingly, as shown in red in Fig 13A, 10 pairs are pseudo symmetric in the parallel interface between GFCC' sheets of VH and VL domains (see also Tables 2 and B in S1 Text for details), with the Q3553-Q3553 hydrogen bond interactions between Glutamine residues present in 99% of the structures, a hallmark of the symmetric pairing of VH and VL domains. This interaction is also present in non-antibody IgV domains such as CD8 dimers (see discussion below). In contrast, the CH1:CL interface exhibits a network of 26 interacting residue pairs present in at least 70% of the Fabs in both dataset with only 3 interacting pairs (shown in red in the Fig 13B) being pseudo symmetric in the antiparallel interface between ABED sheets (see Tables 3 and C in S1 Text for details). With almost double the number of contacts as compared to the VH:VL interface, the conserved CH1:CL interface points to the constant domains' role in maintaining structural integrity of the antibody architecture.

However, let's note that with 107 structures each, our datasets are rather small in regard to the full set of Fabs whose experimental structure is known. While it is beyond the scope of this paper to cover all known structures, we can nevertheless compare the "common interactome" found here with any one taken at random. For example, if we consider a pair of Fabs that target the viral gp120 envelope glycoprotein that binds CD4, the primary receptor for the HIV-1 antigen VRC01 (PDBid:3NGB) and VRC01gl (PDBid:4JPK) [131], they show significant changes in the VH:VL and CH1:CL interfaces (and conformations) due to somatic hypermutations (SHM). They nonetheless possess 11 common pairs in the VH:VL interface and 19 common pairs in the CH1:CL interface with our two datasets (https://www.ncbi.nlm.nih.gov/Structure/icn3d/share.html?dPKhr5EjYpjGPhK19) (see Tables D and E in S1 Text for details).

Finally, let us note that while the initial goal of the IgStrand numbering, as its name indicates, is the study of positional conservation across Ig-domain tertiary structures and across interacting Ig-Ig domain quaternary structures, it can also be used to describe variable regions such as CDRs, although not for positional conservation per se, but for comparative analysis nonetheless. The Table F in S1 Text describes IgStrand numbering of CDRs for Kabat, Chothia, Martin and IMGT delineations (presented in the introduction section).

## Comparison of IgV-IgV quaternary interfaces beyond antibodies

While many different types of interfaces between Ig-domains are possible and do exist, variable domains tend to bind using their (A')GFCC'(C") sheet using the canonical (parallel) IgV-IgV interface of antibodies and TCRs. Canonical parallel interfaces between IgV domains, similar to VH:VL, are found in many protein pairs. This is the case of many immune cell surface receptors such as CD8a/b interacting in cis (same as parallel). If one compares, for example, a VH:VL interface (PDBid: 7N4I) with a VL:VL interface found in Bence-Jones proteins (PDBid: 1REI) and CD8 interface in CD8aa (PDBid: 1 CD8), the IgStrand numbering identifies 7 residue pairs common to all three interactomes, with the conserved symmetrical contacts (igs# 3553-3553), (igs# 4547-9549) and (igs# 9549-4547). These three symmetric contacts were also present in VH:VL domain interactomes in Fabs as shown in the previous section, highlighting the strength of IgStrand numbering in enabling such comparisons. Overall, canonical parallel VH:VL, VL:VL and CD8 interfaces can be easily compared using igs# numbers. VH:VL and CD8aa share twelve interacting residue pairs, while VL:VL and CD8aa share eleven interacting residue pairs (https://structure.ncbi.nlm.nih.gov/icn3d/share.html?4SPQw7jDoGU1tZtbA).

An important point to highlight here is that the canonical interfaces of variable domains (Fig 13A) are C2-symmetric for homodimers such as Bence-Jones protein (VL:VL) dimers as well as CD8aa and retain the pseudo-symmetry for heterodimers like VH:VL as in CD8ab. Variable domains' quaternary structures offer tremendous plasticity in interactomes yet they maintain a pseudo-dimeric interface, as we have seen with antibodies. Besides, a diversity of variable domains' quaternary interfaces exist that depart from the canonical arrangement using the GFCC' sheet. In fact a multitude of interfaces exist involving virtually all regions of Ig domains. Let us note, however, that while many cell surface receptors with N-terminal variable domains such as CD8 dimerize in cis with a canonical parallel interface, they can interact in trans using (for many) a variable domains' inverted (antiparallel) interface, also exhibiting a C2-symmetry [132]. The variable domains' inverted interface can even be found in stable VL:VL homodimers [133]. The study of quaternary Ig-Ig paired interfaces involving the diversity of Ig-domains topological variants is well beyond the scope of this study, but the IgStrand numbering will enable a positional residue-aligned survey all known Ig-Ig interfaces in the protein data bank (PDB). In the following sections, we highlight a number of quaternary as well as tertiary pairings to exemplify the use of the IgStrand numbering.

The Nectin family also presents this parallel interface while interacting in trans [14,119] as well as with cell surface receptors TIGIT [134] and CD96 [135] (see Fig 14). Although the IgV-IgV interfaces use the same GFCC' sheet to bind in either parallel or antiparallel pseudo-symmetric fashion [132] between cell surface receptors and ligands, they use a different and highly specific code of recognition between their respective out-facing residues. The IgStrand numbering is aimed at capturing similarities and differences at any scale to residue interaction networks in the most intimate details across the universe of Ig-domains, irrespective of their topological variations. One should note that many other Ig-Ig interfaces do exist, even when using one sheet or the other, but it is beyond the scope of this paper to cover all these interface variants. The IgStrand numbering is precisely designed to capture the diversity of Ig-Ig interfaces as it can capture the diversity of Ig-fold variants themselves. The IgStrand numbering scheme opens the door to positional structural bioinformatics of Ig domains across one or more predicted structural proteomes. Additionally, for known assemblies in the PDB database today and increasingly available through cryo electron tomography, and hopefully through accurate predictions in the future, the IgStrand scheme enables Ig-interactomes analysis.

## Comparison of more complex Ig-chains interfaces

**Pseudo-quaternary structures in multi-domain chains.** Ig domains can assemble in tandem to form short to long Ig-chains in, mostly, a head to tail arrangement. As mentioned earlier, cell surface receptors containing Ig-domains represent the most populous proteins in the human surfaceome [55–57]. The most well-known are of course the antibodies with chains composed of an IgV domain at the N terminus followed by one (light chains, TCR chains) or more IgC1 domains (heavy chains) forming both heterotypic (light/heavy chains) or homotypic (heavy/heavy) interfaces for CH domains following CH1 and even in CH1 domains in Sharks' heavy chain only antibodies. There are also a multitude of cell surface receptors in the immune system such as B7 receptors and their ligands, or in the nervous system. Ig-chains can contain a diversity of Ig isotypes, mainly IgV, IgC1, IgC2, IgI, FN3 or as a separate type, cadherins. These *isotypes* tend to compose chains in well-defined sequential sets. For example, in vertebrate immune receptors we find numerous chains with an N-terminus IgV followed by one or more IgC1 domains. IgVs can also be followed by C2 domains. Many cell surface receptors also contain a number of FN3 domains, usually after a set of IgV and IgC domains in the membrane proximal regions. The chain lengths encountered can range from a single IgV domain, a couple IgV-IgC domains, or up to 16 or so IgV, IgC, and FN3 domains.

Multiple cell surface adhesion receptors are formed from chains of IgI domains.These chains can be very long, from tens to hundreds of domains, as in Titin [115]. Although we do not have many structures of long chains available in the PDB, the few available exhibit structural plasticity. It is certain that most long chains offer tremendous plasticity with dynamic flexibility in complexes they form, as in the case of obscurin [136] or the Pom210 chains within the nuclear pore complex [113,114]. In the numerous cell surface receptors these Ig-domains in sequence can exhibit local rigidity or flexibility in the linkers or "hinges" that connect them and their residue contacts, and give a chain both structure and plasticity. Some domains may form specific Ig-intrachain interfaces, similarly to observed in quaternary Ig-domain assemblies. Long chains therefore can present a **pseudo-quaternary** structure. Chain geometries and flexibilities depend on the length of linkers/hinges and domain contacts. We do not have experimental structures of long Ig-chains but structure predictions of proteomes [137] may possibly give us sequential Ig-domains pairing, since these tertiary intrachain Ig-Ig interfaces have coevolved within a protein chain.

**Example of Ig-chains forming intrachain pseudo-quaternary Ig-Ig interfaces.** IgV domains tend to assemble using their GFCC' sheets in quaternary interfaces. It is most commonly observed in the PDB as we have seen within antibodies but also in so many cell surface receptors between their N-terminal IgV domains, but this is not the only mode of interaction. The formation of IgV-IgV interfaces depends on the linker length and flexibility between Ig domains. This is obvious as a short linker geometrically forbids chained domain dimerization, long vs. short linkers are commonly used in scFv construction that artificially link IgVH and IgVL domains, where short linkers, between 3 and 5 residues, sometimes called Winter linkers [138] inhibit intrachain dimer formation to favor interchain dimeric formation between VH and VL domains. Experimentally known natural structures involving IgV domains chained in tandem forming IgV-IgV interfaces are not common in known structures of Ig-chains. Two examples of N-terminal double IgV domains, forming a superdomain with an intrachain IgV-IgV interface resembling quaternary interfaces are presented in Fig 15A for CD226 [139] and in Fig 15C for Vcbp3 [140]. The four N-terminal Ig domains are arranged in a highly stable horseshoe conformation resulting from an anti-parallel interaction between domains Ig1–Ig4 and Ig2–Ig3, made possible through a long flexible linker between Ig2 and

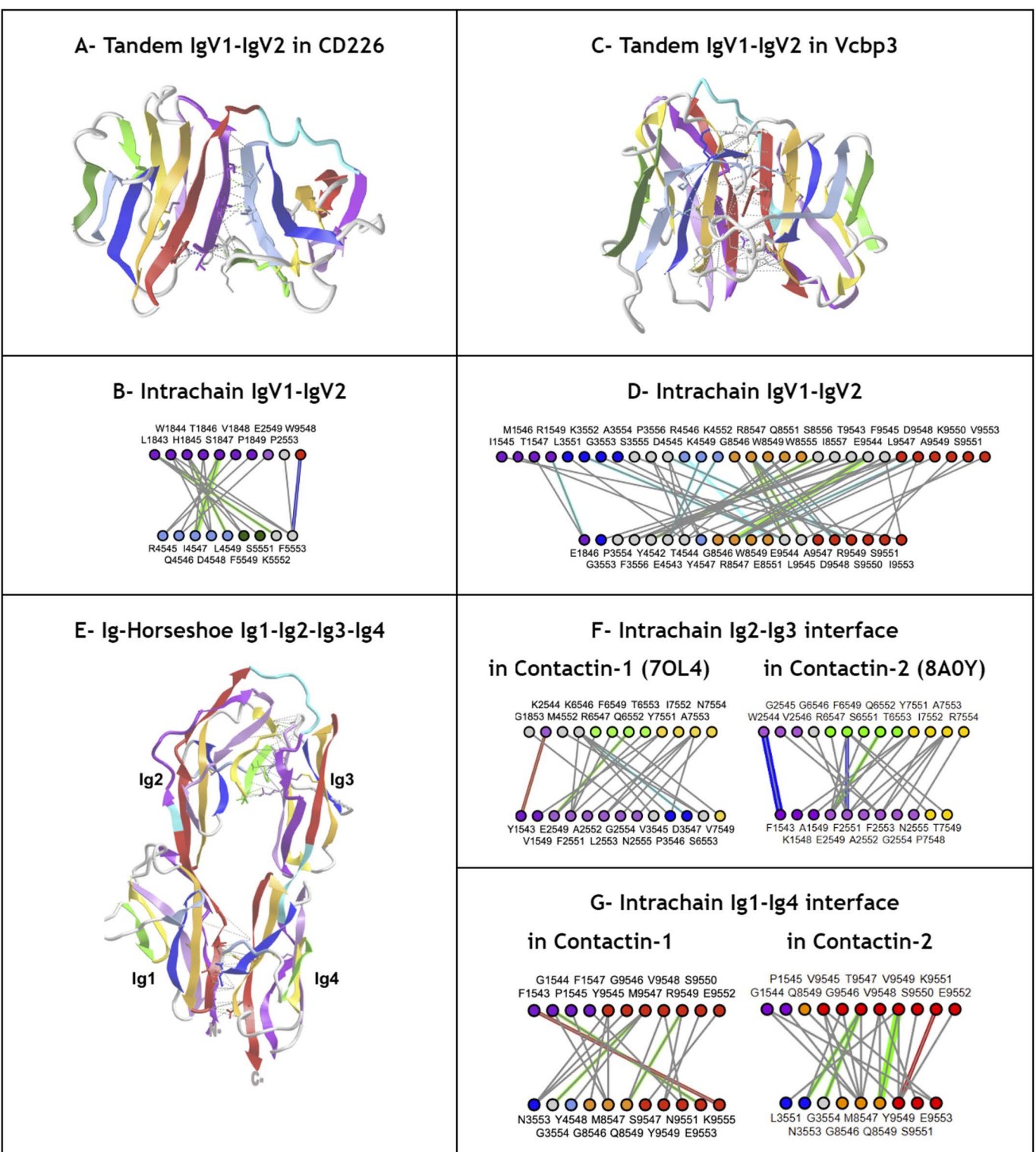

**Fig 15. Multi Ig-domains Intrachain pseudo quaternary interfaces. A-B) Tandem IgV domains in human CD226 (DNAM-1)**. (PDBid 6ISB) The two IgV domains in tandem with a long linker form an antiparallel interface involving a (tertiary) antiparallel strand zipper between IgV1 (strand A') and IgV2 (strand C') as well as a few residues in the G and C" resp. https://structure.ncbi.nlm.nih.gov/icn3d/share.html?UM2jEBErouevRpGR8&t=6ISB. **C-D) Tandem IgV domains of Vcbp3.** (PDBid 2FBO) The variable region-containing chitin-binding protein-3 (VCBP) is an immune-type molecule found in amphioxus (Branchiostoma floridae). The two IgV domains in tandem form an antiparallel interface involving the GFCC' sheets exhibiting C2 pseudosymmetry, partially resembling an

inverted IgV-IgV quaternary interface https://structure.ncbi.nlm.nih.gov/icn3d/share.html?kyVRzncH2ZGzKbQL9&t=2FBO **E-F-G) N-terminal Ig-Horseshoe domain** formed by 4 IgI domains in tandem with a long linker between Ig2 and Ig3 forming an interface Ig2-Ig3 using partially the ABED sheets and an Ig1-Ig4 interface using in part the GFC sheet and the A strand. The 4 domains are highly superimposable (RMSD 4.20/TM-0.77 using TM-Align) when comparing structures of contactin-1 (PDBid 7OL4) and contactin-2 (PDBid 8A0Y) https://structure.ncbi.nlm.nih.gov/icn3d/share.html?iyZJT4PXqFaWnQhP8&t=7OL4,8A0Y.

Ig3 (see Fig 15E). When compared across all diverse proteins, the horseshoe superdomain presents a degree of plasticity (see Fig 15E and 15F comparing Contactin-1 to Contactin-2, and Fig C in S1 Text comparing to DSCAM). The Ig1-Ig4 domain interface shows a significant degree of structural and interactome conservation and may be the main driver in the horseshoe superdomain folding.

**Multi Ig-domains intrachain and interchain quaternary interfaces.** Multiple adhesion protein extracellular chains orchestrate cell-cell communication using a variety of Ig domains, FN3 domains, or cadherin domains, all sharing the Ig-fold [14,123]. For these proteins, cell adhesion equates to Ig-chains binding. Sidekick proteins [141], contactin-2, also called Axonin-1/TAG-1 [142], and other Ig-like molecules form a horseshoe "superdomain" with the four N-terminal Ig domains of the extracellular Ig-chain. These 4-Ig horseshoe superdomains self-interact in either homophilic or heterophilic modes. One mode of quaternary interaction is using a G-strand zipper mechanism between the Ig2 domains, as seen in a structure of contactin-2 in homophilic contact [143,144] and a structure of contactin-1 in heterophilic contact with neurofascin-155 [145] (Fig 16).

Other interfaces besides G-strand zippers also exist between horseshoe superdomains [143,144]. The case of Drosophila DSCAM (Down syndrome cell adhesion molecule) is particularly interesting in this regard, as a very large family of immunoglobulin Ig-protein produced through alternative splicing that play an important role in neural development in wiring the nervous system [120,121]. DSCAMs use a horseshoe superdomain at the N-terminus (Ig1-4) to form a first half-S shape structure using the double Ig1-Ig4 and Ig2-Ig3 intradomain interfaces, and a second half-S shaped through a third Ig5-Ig6 intrachain interface [146]. This S-shaped structure of the 8 N-terminal Ig-domains gives the chain the ability to self-interact through three domains: Ig2, Ig3 and Ig7 simultaneously. To add to the exquisiteness of the homophilic quaternary interface Ig2-Ig2, Ig3-Ig3, and Ig7-Ig7, Nature orchestrates alternative splicing of exons producing Ig2 and Ig3, and Ig7 domains to specifically recognize self- from non-self and assemble accordingly. The three Ig2-Ig2, Ig3-Ig3 and Ig7-Ig7 domain interfaces form distinct antiparallel Ig-Ig interfaces and lie on their common C2-symmetry axis [146] (see Fig 17A). Fig 17B, 17C, and 17D dissect the interactomes of the homophilic quaternary recognition of the three domain pairs.

## Methods & datasets

### Ig-domains nomenclature and coloring scheme

The classical Ig domain nomenclature defines 7 to 9 strands for IgV/I/C1/C2 domains named alphabetically for the 7-strands constant domain IgC1 and adds C' and C" for 9-strands variable domain IgV found in antibodies [46,48,88], The 7-strand IgC2 domain loses the D strand but gains a C' strand w.r.t the IgC1 domain. This can be also seen as a swap of a strand D in sheet ABED to the GFCC' sheet, while the 8-strands IgI domain possesses both C' and D strands. This is how currently the classical four domains have been described and in the following we keep the strand letter names as nomenclature. We use a 7/9 rainbow color spectrum for visualization of strands, from violet to red (Fig 18) for visualization.

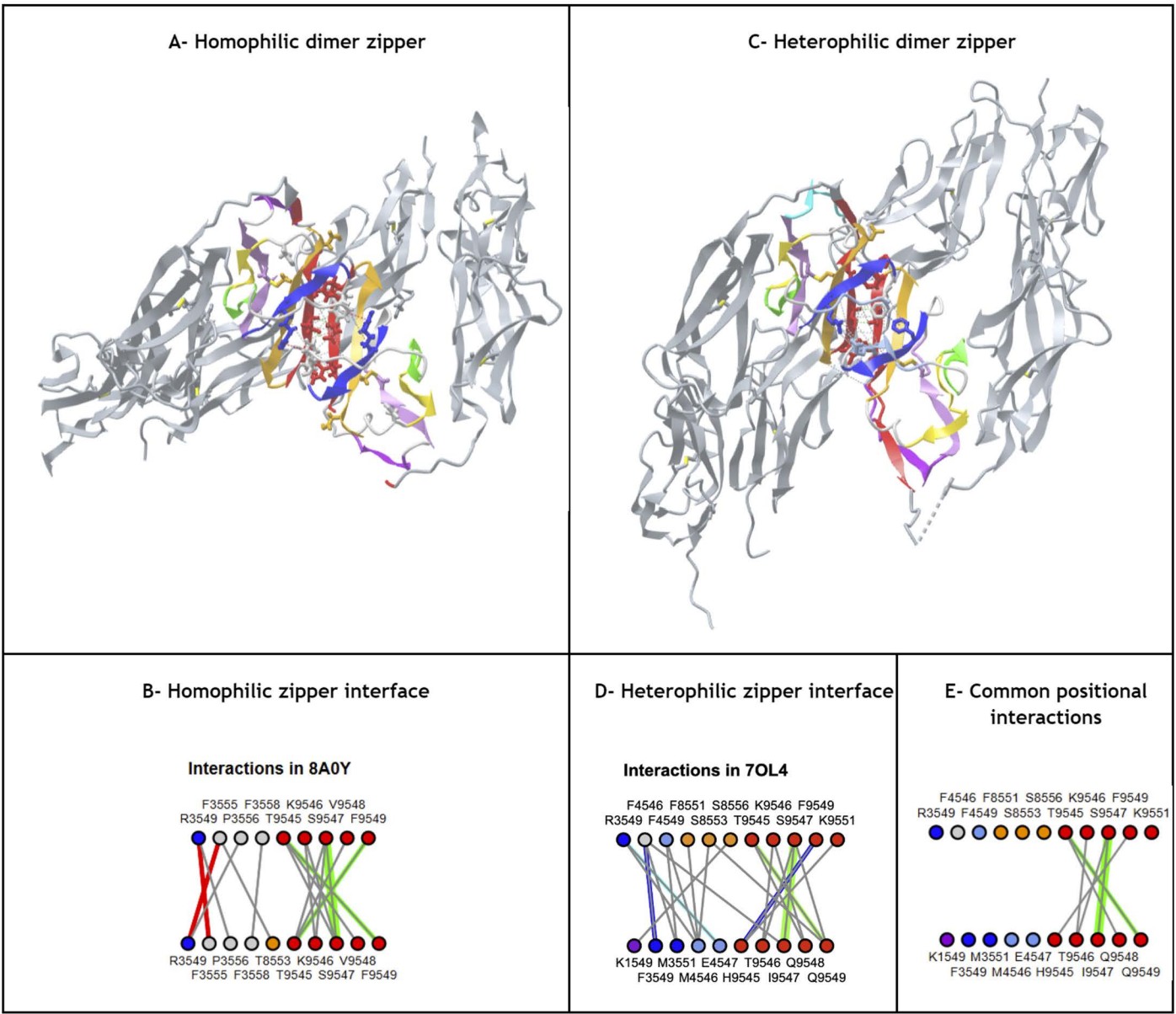

**Fig 16. Homophilic vs. Heterophilic Ig-Horseshoe Zipper interface. A) Contactin-2 Homophilic dimer interface** (PDBid 8a0y). Contactin-2 extracellular region contains 6 N-terminal Ig domains followed by 4 FN3 domains. The four N-terminus Ig-domains Ig1-4 form the Ig-Horseshoe. The Ig2 domain forms a dimer through a G-strand antiparallel zipper interface. **B) Contactin-2 Homophilic Horseshoe Zipper interactome** showing the G-strand zipper (red) as well as residues in the F and C strands of the GFCC' sheet **C) Contactin-1 - neurofascin-155 Heterophilic dimer interface** (PDBid 7OL4). Contactin-1 is highly homologous to Contactin-2 and interacts with **neurofascin-155. Both interact through** their N-terminal Ig-Horseshoe substructure, also using the Ig2 domain forming a G-strand antiparallel zipper interface. The dimer complexes show nonetheless significant plasticity https://structure.ncbi.nlm.nih.gov/icn3d/share.html?ZnVaH-3FLKjYL4Eod8&t=7OL4,8A0Y. **D) Contactin-1 - neurofascin-155 Heterophilic Horseshoe Zipper interactome** showing the G-strand zipper (red) as well as residues in the F, C and C strands of the GFCC' sheet. **E) Conserved positional interaction.** The G-strand zipper interface is conserved in terms of its positional residue network. However One should note that this does not mean that all the atom level residue interactions are the same. For example while the symmetric pair igs# 9547-9547 involves a conserved antiparallel backbone-backbone interaction in both the homophilic (S9547-S9547) or heterophilic (S9547-I9547) pair, the symmetric pair T9545-F9549 in the homophilic dimer (PDBid 8A0Y) forms a side chain(T)-backbone(F) HBond, replaced in the heterophilic dimer (PDBid 7OL4) by a backbone-backbone HBond in the T9545-Q9549 pair while the pseudo symmetric H9545-F9549 pair is reduced to a vdW interaction.

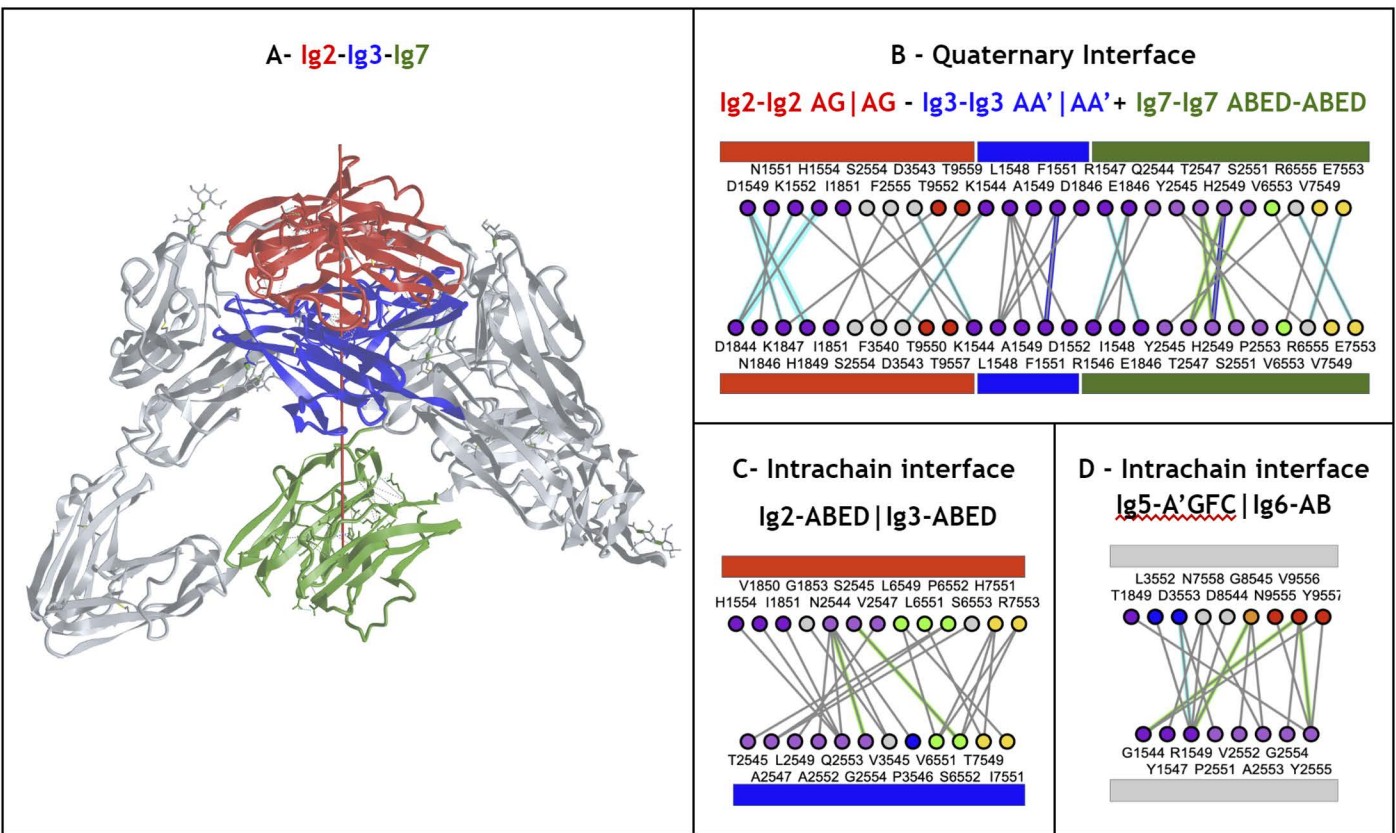

**Fig 17.  A)** https://structure.ncbi.nlm.nih.gov/icn3d/share.html?6swEjWBjXWhufUycA&t=3DMK. **B)** Ig2-Ig2, Ig3-Ig3 and Ig7-Ig7 homophilic interfaces using the AG strands, AA' strands and ABED sheet, respectively. **C)** Ig2-Ig3 intrachain Horseshoe interface using the ABED Sheet. **D)** Ig5-Ig6 intrachain Horseshoe interface using the A'GFC Sheet in Ig6 vs. the A and B strands in Ig6.

### iCn3D Ig domain detection

iCn3D [147,148], pronounced "I see in 3D" is an open-source web-based software program to visualize and analyze molecular structures and interactions. The majority of pictures in this paper and all 3D links are from iCn3D. These links allow the 3D visualization of the proteins described in the paper and their interactions. They integrate the software itself so that the reader can himself or herself visualize, analyze, compare the proteins and their interactions in any way desired. The Ig detection, labeling and coloring algorithm using the IgStrand numbering, termed in short "IgStrand algorithm" is briefly described below and can be used on any protein whose structure is known or predicted.

### Set up the Ig topo-structural variant templates

We use 55 Ig-containing structures as Ig templates to define the strands and loops (see Table 1). The strands in the templates are pairwise aligned using TM-align and the output TM-scores are used to cluster with MEGA11 (Fig D in S1 Text). We grouped the templates into 16 clusters from the clustering. Each template was assigned as one of the following Ig types: IgV, IgC1, IgC2, IgI, IgE, IgFN3, IgFN3-like, and other Ig. Each Ig-domain template (Table 1) has assigned reference numbers for the strands. Fig 18 shows a simplified Ig strand diagram with the central "anchor" residues on each strand labeled. In each strand, the central "anchor" residue is numbered as ij50. Reference numbers are continuous before and after the

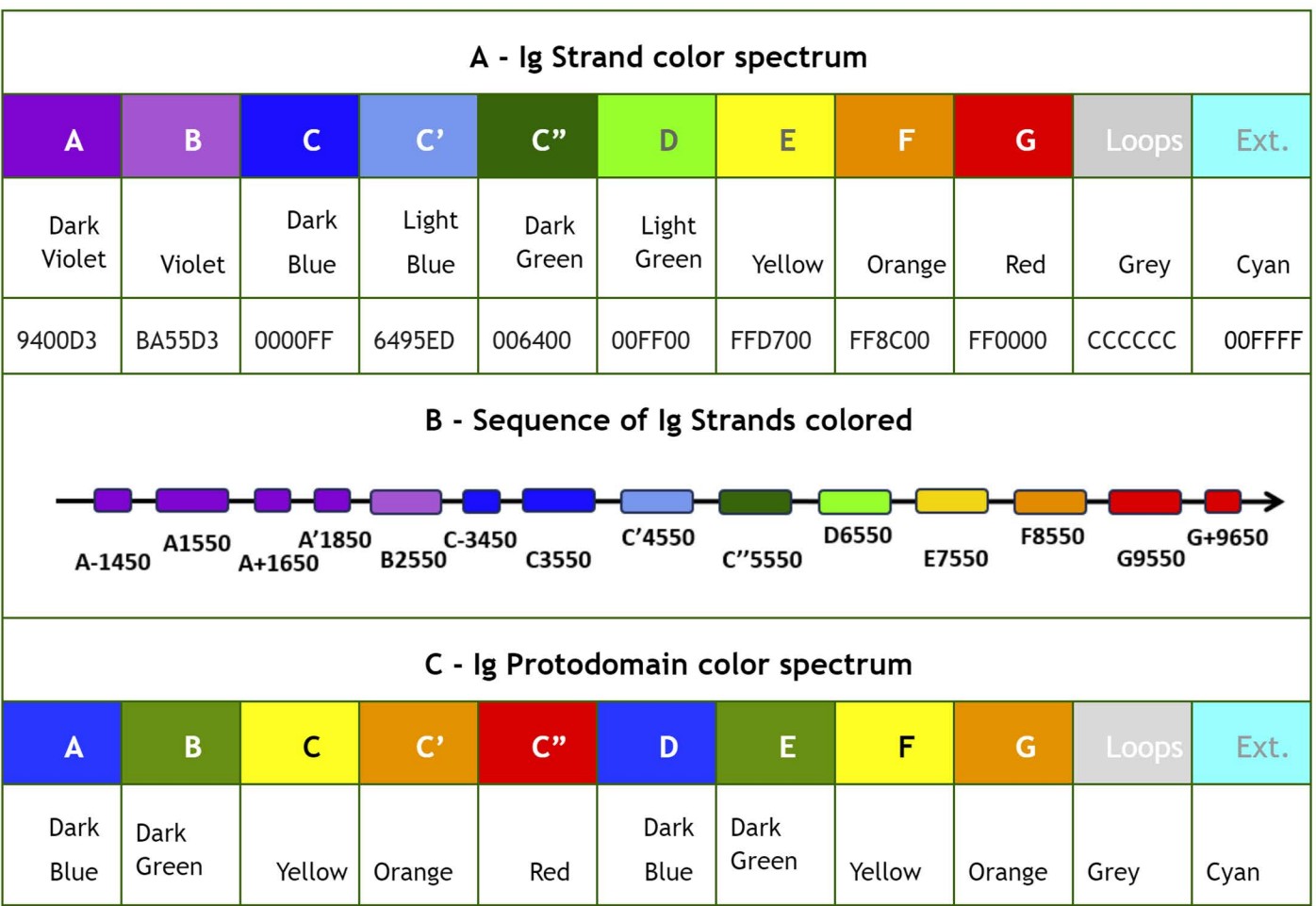

**Fig 18. Ig Strand color spectrum used for ABCC'C"DEFG strands in Ig-domains. A) Ig Strand extended rainbow color spectrum** for 9 strands – 9 colors as indicated. **B) Colored Ig strands according to the Ig Strand Rainbow Spectrum**. Additional and split strands at the N-terminus A',A-,A+,… use the same color as A: dark violet, and similarly additional strands at the C-terminus G+,G++, … use the same color as for G: red. Other inserted strands in Ig-extended domains can appear in white or cyan. **C) Ig Protodomain reduced rainbow color spectrum** 4 strands colors blue, green, yellow, orange for ABCC' and DEFG and a fith color red for the C" strand, if present. Using this color scheme a sheet ABED will be Blue green green blue; a sheet GFCC' will be orange yellow yellow orange. In iCn3D the command used is "color ig strand" and "color ig protodomain" respectively (lowercase). Loops are in Grey between strands. Extensions are in Cyan, and insertions in White. The hexadecimal RGB color codes used in iCn3D are indicated (In this paper, we use the pure yellow code (FFFF00) instead). **Residue Interactions color codes**: Interactomes use the following colors for interaction types **Green**: H-Bonds; **Cyan**: Salt Bridge/Ionic; **Grey**: contacts (Van der Waals); **Magenta**: Halogen Bonds; **Red**: π-Cation; **Blue**: π-Stacking. Contacts are displayed as dotted lines in 3D and traced from Ca to Ca between residues while other types of interactions are atom specific.

anchor, e.g., ij48, ij49, ij50, ij51, ij52, etc., similar to a convention used in GPCRs (Ballesteros and Weinstein 1995). (See main section for details). Each strand is assigned a 1000's number from 1 to 9 for strands A to G, and a color (Fig 18). The conventional strand letter name is also added for convenience in reporting, although redundant with the 1000s digit (see Fig 18 for details). A 100's number is set to 5 for a canonical strand. For example, the anchor of the A strand is #1550, 100's numbers 1-4 and 6–9 are reserved for inserted strands before or after a strand x5xx. For example, A- will be assigned numbers 14xx, A-- 13xx etc., and G+ 96xx, G++ 97xx etc. The A strand can split in IgV and IgI domains: in that case the conventional name A is used for the first half strand on the A-BED sheet with assigned 15xx numbers, and the conventional name A' for the second half strand on the A'GFCC'C" with assigned 18xx numbers.

### Detect Ig domains

We detect Ig domains using structural alignment with the TM-align program [149]. We first split each chain of the input structure into 3D domains, each of which then goes through two steps of alignment to find the best template. The first step is to align the 3D domain with each representative of the 16 clusters to find the top five best alignments. If the best alignment has a TM score larger than 0.85, the templates in that cluster are chosen for the second step alignment. Otherwise, all templates in the top five clusters are chosen. The second step is to align the 3D domain with the chosen templates to find the template with the highest TM score. If the highest TM score is larger than 0.4, an Ig domain is detected in the 3D domain. The two-step alignment is more efficient than the one-step alignment to align each of the 55 templates since the number of total templates may increase in the future. After an Ig domain is detected in a protein chain, the rest of residues in the protein chain iteratively detect additional Ig domains, until no more Ig domains are detected.

### Assign Ig reference numbers

When the detected Ig domain is aligned with the best template, key residues, i.e., "anchors" (see above), are matched to the template. The matched anchor residues are assigned with the reference numbers ij50 (e.g., 1550 for the anchor residue of strand A). The reference numbers in the beta-sheet are then numbered continuously before and after the anchor residue. For coils/loops between two Ig strands, the residues in the first half of the coil are assigned reference numbers continuously from the preceding strand, residues in the second half of the coil are assigned reference numbers continuously to the following strand. If a residue is not in an Ig domain, its reference number is undefined and it will be colored in cyan. Strand insertions are numbered continuously. A strand anchoring mechanism for inserted strands in Ig-extended domains is being developed for future releases.

### How to use iCn3D to assign Ig reference numbers

Once you load structures into iCn3D at https://www.ncbi.nlm.nih.gov/Structure/icn3d/, you can assign IgStrand reference numbers via the menu "Analysis> Ref. Number> Show Ig for Selection", or via the menu "Analysis> Seq. & Annotations" and click the checkbox "Ig Domains" in the "Annotations" section. The reference numbers can be exported via the menu "File> Save File> Reference Numbers".

To process a list of structures, you can download iCn3D Node.js script "refnum.js" at https://github.com/ncbi/icn3d/tree/master/icn3dnode/refnum.js. This script runs TM-align locally using the program at https://github.com/ncbi/icn3d/tree/master/icn3dnode/tma-lign-icn3dnode and the templates at https://github.com/ncbi/icn3d/tree/master/icn3dnode/refpdb. An interaction analysis, as in the "Canonical quaternary antibody IgVH-IgVL and IgCH1-IgCL interactomes section", can be performed via the iCn3D Node.js script "interaction.js" https://github.com/ncbi/icn3d/tree/master/icn3dnode/interaction.js. Both scripts are also available as supplemental files S1 Code and S2 Code.

In setting up packages to run a Node.js script as described at https://github.com/ncbi/icn3d/tree/master/icn3dnode, you can install a specific iCn3D version, i.e., use"npm install icn3d@3.40.0" instead of the "npm install icn3d" instruction that would always use the current version of the program at any particular time.

### Conclusion

The importance of the Ig-fold cannot be overstated. The first human genome project identified the Ig-fold as the most populous in human genes. The proteins coded by these genes

can be found in any region of the cell from the nucleus to the cytoplasm to the extracellular regions of transmembrane proteins. Indeed, a very large number of genes code for cell surface proteins involved in cell adhesion (CAMs) and the development and function of the nervous system, the immune system, the vascular system and the muscular system. This systemic aspect is of particular interest: the Ig-fold in its diverse variants is a **self-interactor** and, precisely, these "systems" use Ig-domains as essential molecular building blocks for bilateral cell recognition with high precision, synapse formation, and communication.

The IgStrand numbering scheme enables **positional comparisons** at the tertiary and quaternary level. Any position in an Ig- domain involved in folding or protein binding can be analyzed in multiple contexts. The IgStrand universal numbering scheme can therefore be used in **comparative analysis** of protein domains exhibiting an Ig-fold and protein chains containing multiple Ig-domains (Ig-chains) as well as Ig-Ig domain interfaces, at any scale. It is suited to study the residues responsible for forming the fold to better understand folding of Ig-domains across the protein universe exhibiting a beta sandwich architecture. A preliminary survey of the human structural proteome using the IgStrand algorithm, considering Ig-, Ig-like and Ig-extended domains, leads to higher estimates of human genes containing Ig-fold domains compared to the 2–3% as originally estimated [1]. The study of the Ig-proteomes across the increasing number of related species' proteomes may also lead to a better understanding of the evolution of genes containing the Ig-fold, especially in the development of the nervous system and immune system.

The formation of synapses through structurally similar cell surface receptors and ligands points to molecular recognition surfaces encoded in Ig-domains surface residues. The IgStrand scheme can be used to precisely study residues facing out and possibly decode Ig-Ig interactomes. As mentioned above, an initial survey of the PDB database leads to a dataset of over 25,000 Ig-Ig domain interfaces that can now all be analyzed in parallel to better understand both the folds and their molecular interactions, towards an understanding of co-evolution of receptors ligand pairs and their codes on recognition (work in progress).

We expect the parallel analysis of protein families containing Ig-, Ig-like, or Ig-extended domains to reveal potential clues in biology, not just on evolution that led to complex multicellular organisms with an immune system, a nervous system, a vascular system, and a muscular system, but on protein folding. Beyond research on evolution, folding, and molecular assemblies involving the Ig-fold, the IgStrand scheme can be used in assisting protein engineering, and we can foresee a number of direct applications for example in antibody and single Ig-domain (nanobody) design, and CAR T-cell therapies, especially when when both target binding moieties and antigen targets themselves consist of Ig-domains or Ig-chains. Indeed, it is important to realize that a multitude of antigen targets in immunology are extracellular Ig-chains with one, two (CD19) or more (CD22) of single pass transmembrane receptors [11,150].

We have implemented the IgStrand numbering scheme in iCn3D [147] to enable interactive analysis. An Ig-fold recognition algorithm based on IgStrand enables the accurate identification of Ig-domains across any protein structure. We refer to it in short as the IgStrand algorithm. The software can also be used in batch mode [148], enabling large datasets analyses. As mentioned above an initial survey of over 20,000 genes and 82,518 proteins of the structural human proteome [71,137] indicates the presence of the Ig-fold in a much larger proportion than the original estimates. Improvements are still needed to accurately label each and every type of topo-structural variant. Once we reach that stage, we can envision the study of evolution of all Ig-domains and proteins across structural proteomes.

The Ig-fold is unique in its ability to sustain topological variations and we have focused on developing a universal numbering scheme that can likewise sustain these variations. We

deliberately chose to make the IgStrand numbering human as well as machine readable as interactive analysis is central to our approach. In the future we envision extensions of IgStrand with a hierarchical numbering, to manage long Ig-chains and their interactions, as well as explicit alternative positioning of strands in beta-sandwich sheets to extend the nomenclature to jelly rolls and other beta-sandwich topologies. Finally, loop regions present challenges in universal numbering that need further innovation. The IgStrand scheme was a necessary step towards positional structural bioinformatics of Ig-proteomes and Ig-interactomes, yet other important folds would benefit from the development of their own universal reference numbering, while offering less topological challenges, to provide a means to full scale positional structural bioinformatics.

## Supporting information

**S1 Text.** **Fig A. CDD IgSF Tree**. Constructed with BLOSUM 62 with the Score of Optimally-Extended Block method. **Fig B.** F strand Tyr (igs# 8548 or 8546) is highly conserved across many eukaryotic and bacterial Ig-like domains despite topo-structural variations in the fold. This Tyr appears to play similar roles by interacting with the EF loop, and may provide insights into the evolution of a number of proteins sharing the Ig-fold. It certainly points toward a key structural residue. **Fig C.** Tertiary Intrachain interfaces in the Horseshoe superdomain formed by the four N-terminal residues in Ig-chains. This superdomain allows a certain level of plasticity observed in comparing contactin-2 and DSCAM. The RMSD is 4.9Å. A) Contactin-2 Horseshoe (Pdbid 8A0Y) shows an Ig1-Ig4 and a Ig2-Ig3 antiparallel interfaces. B) Ig1-Ig4 interactome in contactin-2. C) Ig2-Ig3 interactome in contactin-2. D) Ig1-Ig2 interactome in contactin-2. E) Ig3-Ig4 interactome in contactin-2. F) Ig1-Ig4 interactome in DSCAM. G) Ig2-Ig3 interactome in DSCAM. H) Ig1-Ig2 interactome in DSCAM. I) Ig3-Ig4 interactome in DSCAM. J) Common interactions in Ig1-Ig4. This pairwise interactome is more conserved than Ig2-Ig3 that is more plastic. K) Common interactions in Ig3-Ig4. **Fig D.** Clustering Ig templates using MEGA11. The graph shows the distances between templates. The corresponding TM-score is equal to 1 - distance. **Table A.** Number of Heavy chain - Light chain contacts in VH:VL and CH1:CL interfaces in Fabs bound to SARS-CoV2 antigens or to a diverse set of antigens and present to more than 70% (or 90%) of the Fabs in each dataset. **Table B.** VH:VL interactions of Fabs binding diverse antigens (70% cutoff). Red numbers represent symmetric contacts. Shaded cells represent five highly conserved contacts (90% cutoff) shared between the SARS-CoV-2 antigen binding dataset (Table 2 in manuscript) and this diverse antigen binding dataset. Bold contacts represent highly conserved hydrogen bonding contacts. Underlined contacts represent those that have been found previously [131]. **Table C.** CH1:CL interactions of Fabs binding diverse antigens (70% cutoff). Red numbers represent symmetric contacts. Shaded cells represent twelve highly conserved contacts (90% cutoff) shared between SARS-CoV-2 antigen binding dataset (Table 3) and this diverse antigen binding dataset. Bold contacts show highly conserved hydrogen bonding and in purple for a highly conserved ionic contact. **Table D.** VH:VL common interaction pairs among Fabs binding the SARS-CoV-2 spike protein, Fabs binding a diverse set of antigens, and a pair of Fabs (PDB IDs: 3NGB, 4JPK) targeting the viral gp120 envelope glycoprotein. Red numbers represent symmetric contacts. **Table E.** CH1:CL common interaction pairs among Fabs binding the SARS-CoV-2 spike protein, Fabs binding a diverse set of antigens, and a pair of Fabs (PDB IDs: 3NGB, 4JPK) targeting the viral gp120 envelope glycoprotein. Red numbers represent symmetric contacts. **Table F.** CDR regions as defined by previous Ig numbering systems, mapped to IgStrand numbers.
(DOCX)

**S1 Data. Multiple Sequence Alignment of IgV domains in fasta format.**
(FASTA)

**S2 Data. Multiple Sequence Alignment of IgI domains in fasta format.**
(FASTA)

**S3 Data. Multiple Sequence Alignment of IgC1 domains in fasta format.**
(FASTA)

**S4 Data. Multiple Sequence Alignment of IgC2 domains in fasta format.**
(FASTA)

**S5 Data. Multiple Sequence Alignment of IgFN3 domains in fasta format.**
(FASTA)

**S6 Data. Multiple Sequence Alignment of IgCad (Cadherin) domains in fasta format.**
(FASTA)

**S7 Data. Ig-domains templates of Table 1 in Excel format.**
(XLXS)

**S8 Data. IgStrand Alignment of 107 SARS-CoV-2 antigen binding VH, VL, CH1, CL domains in Excel format.**
(XLXS)

**S9 Data. IgStrand Alignment of 107 diverse antigen binding VH, VL, CH1, CL domains in Excel format.**
(XLXS)

**S1 Code. This is the refnum.js script from iCn3D node.js framework, to assign IgStrand numbers to any input protein structures with Ig domains.**
(JS)

**S2 Code. This is the interaction.js script from iCn3D node.js framework, to calculate protein-protein interactions across any quaternary interface.**
(JS)

## Acknowledgments

The authors thank Dr. Aron Marchler-Bauer from the National Center for Biotechnology Information, National Library of Medicine for useful discussions.

## Author contributions

**Conceptualization:** Ravinder Abrol, Philippe Youkharibache.

**Data curation:** Caesar Tawfeeq, Jiyao Wang, Umesh Khaniya, Thomas Madej, James Song, Ravinder Abrol, Philippe Youkharibache.

**Formal analysis:** Caesar Tawfeeq, Umesh Khaniya, Thomas Madej, James Song, Ravinder Abrol, Philippe Youkharibache.

**Funding acquisition:** Ravinder Abrol, Philippe Youkharibache.

**Investigation:** Caesar Tawfeeq, Jiyao Wang, Thomas Madej, Ravinder Abrol, Philippe Youkharibache.

**Methodology:** Caesar Tawfeeq, Jiyao Wang, Thomas Madej, Ravinder Abrol, Philippe Youkharibache.

**Project administration:** Ravinder Abrol, Philippe Youkharibache.

**Resources:** Jiyao Wang, Ravinder Abrol, Philippe Youkharibache.

**Software:** Jiyao Wang, Umesh Khaniya, Thomas Madej, Philippe Youkharibache.

**Supervision:** Ravinder Abrol, Philippe Youkharibache.

**Validation:** Ravinder Abrol, Philippe Youkharibache.

**Visualization:** Caesar Tawfeeq, Thomas Madej, James Song, Philippe Youkharibache.

**Writing – original draft:** Philippe Youkharibache.

**Writing – review & editing:** Caesar Tawfeeq, Jiyao Wang, Umesh Khaniya, Thomas Madej, James Song, Ravinder Abrol, Philippe Youkharibache.

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
