## [Decision Letter · Decision Letter 0]

11 Nov 2024

PCOMPBIOL-D-24-01553A universal residue numbering scheme for the Immunoglobulin-fold (Ig-fold) to study Ig-Proteomes and Ig-InteractomesPLOS Computational Biology Dear Dr. Youkharibache, Thank you for submitting your manuscript to PLOS Computational Biology. After careful consideration, we feel that it has merit but does not fully meet PLOS Computational Biology's publication criteria as it currently stands. Therefore, we invite you to submit a revised version of the manuscript that addresses the points raised during the review process.

 Please submit your revised manuscript within 60 days Jan 11 2025 11:59PM. If you will need more time than this to complete your revisions, please reply to this message or contact the journal office at ploscompbiol@plos.org. Please include the following items when submitting your revised manuscript: * A rebuttal letter that responds to each point raised by the editor and reviewer(s). You should upload this letter as a separate file labeled 'Response to Reviewers'. This file does not need to include responses to formatting updates and technical items listed in the 'Journal Requirements' section below.* A marked-up copy of your manuscript that highlights changes made to the original version. You should upload this as a separate file labeled 'Revised Manuscript with Track Changes'.* An unmarked version of your revised paper without tracked changes. You should upload this as a separate file labeled 'Manuscript'. If you would like to make changes to your financial disclosure, competing interests statement, or data availability statement, please make these updates within the submission form at the time of resubmission. Guidelines for resubmitting your figure files are available below the reviewer comments at the end of this letter. We look forward to receiving your revised manuscript. Kind regards, Iddo Friedberg, Ph.D.Academic EditorPLOS Computational Biology Arne ElofssonSection EditorPLOS Computational Biology Feilim Mac GabhannEditor-in-ChiefPLOS Computational Biology Jason PapinEditor-in-ChiefPLOS Computational Biology  **Journal Requirements:** **Additional Editor Comments (if provided):****Reviewers' comments:** Reviewer's Responses to Questions

**Comments to the Authors:**

Reviewer #1: The authors provide an updated numbering scheme to identify Ig structural proteomes and interactomes, which allow researchers to communicate about protein structures and function in a standardized fashion. The numbering scheme is innovative in that it does not solely rely on sequence alignments but incorporates the structure and folding patterns. This new scheme is human readable as well as machine readable, allowing for new machine learning algorithms to be implemented.

Comments:

1. The Introduction is a nice review on Ig folds but doesn’t mention the previous numbering schemes until the Results and Discussion section. The information becomes a bit redundant. I would suggest moving some of the discussion in the “Results and Discussion” to the Introduction section to differentiate between older numbering schemes and the contribution of this current work. The “Definition of the IgStrand numbering” section on page 15 seems to be the start of the Results section.

2. Terms are defined later in the article after they are used instead of the other way around. An example is the description of the A strand split on page 12, when it is mentioned on page 6.

3. Page 4, last paragraph, the second sentence is a long incomplete sentence with several references making it difficult to read.

4. Page 15, the scheme is described as using 4 digits, but the 3rd and 4th digit are not discussed until a couple of sections later. A brief mention of all four in the beginning would help with the transitions.

5. Page 6, last paragraph, first sentence: “The immunoglobulin fold is both one and many” is a bit confusing.

6. Figure 3 legend. What does the # represent in the figure? CD4, ICOS, PD-L2, and VNAR domains are mentioned in the legend but not shown in the figure. This should remain in the main text of the article.

7. Table 1. The font should be bigger. It is difficult to read.

8. Page 30, last paragraph, first sentence: The word “least” is written twice.

Reviewer #2: In this work, authors have developed a universal numbering system IgStrand numbering that enables positional comparison of the Immunoglobulin fold (Ig-fold). Ig-fold belong to the Ig Superfamily and is the most populous domain in human genome accounting for 2% of the coding genes. Ig domains are ubiquitous, playing important roles in cellular communications and Immunoglobulin variable domains are the most studied Ig domains.

Ig domains are made of 7-9 strands that make a sandwich of two β sheets. IgStrand numbering (ij50) consists of 4 digits and assigns number 50 to the “anchor residue” in each of the 9 main canonical Ig domain strands: A, B, C, C’, C’’, D, E, F and G. Anchor residues are assigned based on both structural considerations and sequence conservation. Digit “i” spans from 1-9 and represents the canonical strands with their respective order. Digit “j” is usually 5, but strand insertions and splits can be assigned with j+1 or j-1. Residues are number linearly ij50-n or ij50+n if they come before or after the anchor residue.

Using their IgStrand numbering, authors present diverse set of Ig-folds such as Arrestin, Lamin, Transcription factors and Ig-domains in viruses shining light on their structural diversity and variations. Then authors explore Ig-Ig domain interfaces in library of Fab fragments that target SARS-CoV-2 spike protein identifying 361 + 213 unique interacting residue pairs for VH|VL and CH1|CL respectively. Finally, authors explore more complex Ig-chain interfaces such as Tandem IgV1-IgV2 and Ig-Horseshoe domains.

Comments.

• Authors explore an interesting topic, and they managed to come up with a sequence numbering system for the entire Ig Superfamily.

• Authors were able to present their IgStrand numbering in a clear and easy to digest way with nice visual presentation of anchor residues (Figure 6).

• IgStrand numbering is linear and easy to read for humans making comparison of Ig-folds with numerous topostructural variations easier. Furthermore, machine readability of IgStrand numbering could enable comparison of large dataset of Ig domains and quaternary interfaces.

Critical comments:

• Figure 3 a. b. c. is lowercase, but other figures are capitalized such as Figure 5 A, B, C.

• I liked the fact that Authors made IgStrand numbering linear by reserving j for strand insertions and splits (Figure 18 B). A-, and C- are numbered j = 5-1 and A+ and G+ strands are numbered j = 5+1. Although it was mentioned in Page 16 that A’ is a special case and its anchor was numbered j = 5+3 (1850). However, I did not understand why A strand skips j = 7. Was it reserved for something? I think authors should clarify that.

• In Page 30, Authors compare library of 258 Fab fragments. They found 361 unique interacting pairs for VH|VL and they found 231unique interacting pairs for CH1|CL. Since all of these antibodies target the same protein, it is expected to see common interacting pairs for VH|VL even if they might be targeting different epitopes within the same protein. However, considering CH and CL domains are constant, one could expect more common pairs for CH1|CL relative to VH|VL. Authors should further explore this by comparing VH|VL and CH1|CL in library of random Fabs targeting different antigens (ideally, similar size library) to see if common antigen target increases the common residue pairs in VH|VL.

Overall, Authors present interesting and impactful study that can help other researchers in the field. They give enough background to highlight the novelty in their work and their explanation is easy to digest. However, I believe the experiment suggested above is critical to understand the result of IgVH-IgVL and IgCH1-IgCL interactome experiment. Therefore, I believe Authors should address this question along with other two minor comments before publication.

**Have the authors made all data and (if applicable) computational code underlying the findings in their manuscript fully available?**

Reviewer #1: Yes

Reviewer #2: Yes

PLOS authors have the option to publish the peer review history of their article (what does this mean? ). If published, this will include your full peer review and any attached files.

**Do you want your identity to be public for this peer review?** For information about this choice, including consent withdrawal, please see our Privacy Policy .

Reviewer #1: No

Reviewer #2: No

 **Figure resubmission:**While revising your submission, please upload your figure files to the Preflight Analysis and Conversion Engine (PACE) digital diagnostic tool, https://pacev2.apexcovantage.com/. PACE helps ensure that figures meet PLOS requirements. To use PACE, you must first register as a user. Registration is free. Then, login and navigate to the UPLOAD tab, where you will find detailed instructions on how to use the tool. If you encounter any issues or have any questions when using PACE, please email PLOS at figures@plos.org. Please note that Supporting Information files do not need this step. If there are other versions of figure files still present in your submission file inventory at resubmission, please replace them with the PACE-processed versions. 
---

## [Editor Report · Decision Letter 1]

20 Jan 2025

Dear Dr. Youkharibache,

We are pleased to inform you that your manuscript 'IgStrand: A universal residue numbering scheme for the Immunoglobulin-fold (Ig-fold) to study Ig-Proteomes and Ig-Interactomes' has been provisionally accepted for publication in PLOS Computational Biology.

Best regards,

Iddo Friedberg, Ph.D.

Academic Editor

PLOS Computational Biology

Arne Elofsson

Section Editor

PLOS Computational Biology
